



# High frequency broadband acoustic systems as a tool for high latitude glacial fjord research

Elizabeth Weidner[1], Grant Deane[1], Arnaud Le Boyer[1], Matthew H. Alford[1], Hari Vishnu[2], Mandar Chitre[2,3], M. Dale Stokes[1], Oskar Glowacki[4], Hayden Johnson[1], and Fiammetta Straneo[2]

[1]Scripps Institution of Oceanography, University of California San Diego, San Diego, 92037, USA
[2]Acoustics Research Laboratory, National University of Singapore, Singapore, 119007, Singapore
[3]Department of Electrical and Computer Engineering, National University of Singapore, Singapore, 119007, Singapore
[4]Institute of Geophysics, Polish Academy of Science, Warsaw, 6X7P+87, Poland

*Correspondence to*: Elizabeth Weidner (ereedweidner@ucsd.edu)

**Abstract.** High frequency broadband echosounders enable the monitoring of complex dynamics, through rapid collection of high resolution, near-synoptic observations of the water column and quantitative geophysical measurements. Here, we demonstrate the applicability and utility of broadband active acoustics systems to improve observational capabilities in high latitude glaciate fjords. These isolated and challenging field locations are a critical environment, linking the terminal end of terrestrial ice sheets to the broader ocean, undergoing complex changes due to accelerated high-latitude warming trends. Using broadband (160-240 kHz) acoustic data, collected in tandem with ground truth measurement from a CTD and microstructure probe, in Hornsund fjord in southwest Svalbard we address three crucial topics: 1) variability of the thermohaline structure and mixing across different temporal and spatial scales, 2) identification and characterization of processes in play at dangerous glacier terminus, and 3) remote estimation of dissipation rates associated with mixing. Through these analyses, we illustrate the potential of broadband echosounders as a relatively low-cost, low-effort addition to experimental field kits, well suited for field deployment in high-latitude fjords where observations are limited by length of season and generally challenging conditions.

## 1 Introduction

The ice-water interface of marine-terminating (tidewater) glaciers in high-latitude fjords represents a critical transition zone, linking the terrestrial, cryosphere, and ocean systems. As such, tidewater glaciers are central to not only polar but global ocean dynamics: contributing to sea level rise through ablation (Joughin et al., 2012, 2004; Nick et al., 2009), modifying ocean properties through freshwater flux (Fichefet et al., 2003; Straneo et al., 2011), altering thermohaline circulation patterns (Bamber et al., 2012; Beaird et al., 2023; Böning et al., 2016; Slater et al., 2018a; Straneo and Cenedese, 2015), and supporting highly productive and unique marine ecosystems (Hopwood et al., 2020, 2018; Lydersen et al., 2014). Simultaneously, their direct connection to atmospheric and oceanic forcing, combined with accelerated high-latitude warming trends (England et al., 2021; Holland et al., 2008; Howat et al., 2007; Serreze and Francis, 2006) has resulted in a dramatic increase of glacial retreat



rates and ice mass loss over the past two decades (Enderlin and Hamilton, 2014; Geyman et al., 2022; van den Broeke et al., 2016; IPPC, 2019). Despite their importance to polar oceanography, many questions remain in quantifying the impact of changing climate on tidewater glacier systems and predicting the subsequent impact of these changes on the broader polar system (e.g., Straneo and Heimbach, 2013). Here, we outline the potential of broadband active acoustic systems to help answer

many outstanding questions of these high-latitude coastal regions.

A central challenge in characterizing these regions lies in the gap between the observational requirements, both in terms of sampling rate and spatial scale, and current observational capabilities. High latitude fjords are isolated, often dangerous environments, where the research season is short, the presence of ice and poor weather limits direct observation, and the field conditions can be physically and psychologically demanding (Leon et al., 2011; Palinkas and Suedfeld, 2008). Moreover,

while traditional in-situ measurement techniques, such as conductivity-temperature-depth (CTD) or microstructure profilers, can provide high resolution vertical profiles of geophysical measurements (e.g., temperature, salinity), they lack spatial context and practical deployment mechanisms in many regions of high latitude fjords, e.g., near the ice face. Satellite-based data is commonly used for broad-scale geophysical observations (Konik et al., 2021; Serreze and Stroeve, 2015); however, the spatial resolution of this data can be coarse (>15 m, Sentinel-2, Landsat-9), depending on the data source. Hence observations, which

are often limited by ice and cloud cover, can only provide direct information on the upper meters of the water column (Swift, 1980). These compounded observational challenges often result in qualitative, rather than quantitative, descriptions of critical inner-connected fjord processes, such as the connection between stratification and glacial ablation processes (submarine melting, surface melting, and calving). Enhanced observational capabilities, for both synoptic and in-situ quantitative geophysical measurements, are needed to fill the observational gaps in high latitude fjords and quantify the ongoing impact of

the changing climate.

Active acoustic systems can provide these measurements. They offer near-synoptic observations of water column dynamics along km-scale tracks on length scales of shorter than one meter (Chu and Stanton, 1998). Acoustic profiles can be collected rapidly, typically at a rate of 0.1-10 Hz, providing coverage of spatial scales from the sub-meters to full-fjord (kms), and temporal scales ranging from seconds to seasons (Godø et al., 2014; Proni and Apel, 1975). Oceanographic processes have a

long history of acoustic observation, such as internal waves (Moum et al., 2003; Orr et al., 2000), hydraulic jumps (Cummins et al., 2006; Farmer and Dungan Smith, 1980), stratification (Holbrook et al., 2003; Penrose and Beer, 1981; Ross and Lavery, 2009; Stranne et al., 2017; Weidner et al., 2020), fish and zooplankton communities (Cotter et al., 2021; Foote et al., 2005; Lavery et al., 2010), gas bubbles (Marston et al., 2023; Medwin, 1977; Vagle et al., 2005), suspended sediment (Thorne and Hanes, 2002; Thorne and Hardcastle, 1997; Thorne and Hurther, 2014; Young et al., 1982), and fluid emissions (Bemis et al.,

2012; Xu et al., 2017). The remote nature of acoustic data collection minimizes risks and logistical challenges associated with direct sampling in challenging environments (e.g., Xu et al., 2021). Active acoustic systems have provided high resolution bathymetry of polar seas (Björk et al., 2018; Jakobsson et al., 2020), information on fjord circulation patterns (Abib et al., 2024; Sutherland and Straneo, 2012), and information on sea ice extent (Bourke and Garrett, 1987; McLaren et al., 1994; Rothrock and Wensnahan, 2007); however, they remain underused in the study of the coastal water column. In this manuscript,





we discuss the potential of filling current observational gaps in high latitude glacial fjord systems using high frequency (>100 kHz), broadband, split-beam echosounders.

## 1.1 Overview of split-beam echosounders

Split-beam echosounders were initially developed as narrowband (single frequency) tools for fisheries applications (e.g., Jech and Michaels, 2006; Simmonds and MacLennan, 2008) and have a long history of producing qualitative plots of acoustic
scattering intensity in the water column across both range and depth; see Fig. 7 in Farmer and Armi (1999), Fig. 1 in Moum et al. (2003), and Fig. 3 in Jech and Michaels (2006). These images provide intuitive and compelling qualitative visualization of oceanic processes and boundaries for applications ranging from target identification in recreational fish finders to long-term environmental monitoring by government agencies (e.g., Jech, 2021) to descriptions of hydrological forcing in highly dynamic environments (e.g., Farmer and Armi, 1999). Measurements of absolute acoustic scattering strength from split-beam
echosounders can be converted into quantitative data streams of important geophysical signals through acoustic inversion methods with the proper field calibration procedure (Demer et al., 2015), in-situ sampling, and theoretical acoustic scattering models. Given the potential for rapid, remote measurements of geophysical properties of the ocean interior, there has been significant effort directed towards developing inversion methods, included analysis of turbulent mixing (Goodman, 1990; Moum et al., 2003; Ross and Lueck, 2005; Seim et al., 1995), seafloor sediment characteristics (Amiri-Simkooei et al., 2011;
Fonseca and Mayer, 2007; LeBlanc et al., 1992; McGee, 1990), biomass estimates for fisheries management and community analysis (Martin et al., 1996; Sawada et al., 1993), bubble size distributions (Li et al., 2020), and heat flux from hydrothermal plumes (Xu et al., 2017).

The incorporation of broadband capabilities (signal pulses with an extended and continuous frequency range) has dramatically increased the utility of split-beam systems for the quantitative study of oceanic processes. Compared to narrowband systems,
broadband echosounders provide increased along-beam (range) resolution - on the order of decimeters to millimeters - and continuous frequency resolution for spectral characterization (Chu and Stanton, 1998; Ehrenberg and Torkelson, 2000). Increased resolution is leveraged by researchers to discriminate closely spaced targets in the water column, facilitating precise positioning and tracking of targets (e.g., Loranger and Weber, 2020; Weidner et al., 2019; Jerram et al., 2015); while continuous frequency-modulated scattering over the pulse bandwidth reduces the ambiguities in the interpretation of the acoustic returns
and specific scattering phenomena can be differentiated and more accurately classified in regions where scattering intensity and spatial context alone are not sufficient (e.g., Lavery et al., 2010; Stanton et al., 2010). Spectral analysis has been leveraged for acoustic inversion procedures to quantify biomass and determine the community make-up of both plankton (Foote et al., 2005; Lavery et al., 2010) and fish (Cotter et al., 2021; Loranger et al., 2022), as well as differential between multiple scattering mechanisms, for example oil droplets and gas bubbles (Loranger and Weber, 2020). A growing body of literature discusses
broadband acoustic inversion methods using commercial broadband echosounders to quantify oceanic phenomena including thermohaline structure (Loranger et al., 2022; Weidner and Weber, 2024), turbulent mixing (Lavery et al., 2013; Muchowski et al., 2022), and gas seeps (Weidner et al., 2019).



Until recently the use of broadband spectral analysis methods has been limited in their applicability due to the high cost of custom-made broadband systems, often exceeding $500,000. The availability of lower cost, commercial broadband systems

and the expanded access to a broader user base has resulted in a growing body of literature and processing software for broadband data analysis (Blomberg et al., 2018; Cotter et al., 2019; Demer et al., 2017; Lavery et al., 2017; Weidner et al., 2019). Hence the potential for these systems to improve observational capabilities in isolated and challenging field locations, such as high-latitude fjords. In this manuscript, a high frequency (160-240 kHz), broadband split-beam acoustic dataset, along with direct sampling from a conductivity-temperature-depth (CTD) and microstructure probe, were collected in Hornsund

fjord, Svalbard. This dataset is used to demonstrate the applicability and utility of broadband active acoustics systems to address three crucial topics:

1) Variability of the thermohaline structure and mixing across different temporal and spatial scales,

2) identification and characterization of processes in play at dangerous glacier terminus, and

3) remote geophysical parameter estimation

Section 2 of this manuscript provides an overview of the field site in Svalbard. The equipment, vessel, and data collection methods are described in Sect. 3. Section 4 covers the broadband echosounder data analysis, which includes the use of high-resolution water column observations in tandem with direct sampling to characterize dynamics across the full-fjord scale, as well as the potential for remote measurements in dangerous regions, specifically at the submerged glacial terminus. Section 4 also includes a discussion of broadband spectral analysis and acoustic inversion efforts to remotely measure turbulent mixing associated with fjord bathymetry and tidal forcing. A discussion of the acoustic data collection methods and analysis, including

limitations of broadband systems is covered in Sect. 5. Section 6 concludes the paper with the potential for future applications of broadband systems to high-latitude fjords.

## 2 Study site

Data were collected between July 6th and 12th, 2023 in Hornsund Fjord (Fig. 1). Hornsund is a highly glaciated, ~35 km long

fjord located in southwest Spitsbergen, Svalbard. Glaciers cover approximately 67% (802 km$^2$) of the total area of Hornsund fjord, with tidewater glaciers constitute 97% of its glacierized area (Błaszczyk et al., 2013). The western coast of Svalbard, including Hornsund Fjord, is influenced mainly by the North Atlantic Ocean current (Walczowski and Piechura 2006). The mean annual air temperature trend of the region over the last four decades is more than +1.14 °C decade$^{-1}$, a rate six times higher than the global average (Wawrzyniak and Osuch, 2020). Tidewater glaciers in Hornsund Fjord are retreating at an

average rate of ~70-100 m yr$^{-1}$ (Błaszczyk et al., 2023, 2013), among the highest rates in Svalbard.

Data collection in Hornsund was focused on the glaciated bay of Hansbukta and the calving front of the retreating, grounded tidewater glacier Hansbreen. The bathymetry of Hansbukta is dominated by rapid shallowing from a main channel of Hornsund fjord (~200 m deep) to a well-defined transverse ridge (3-15 m deep), which is interpreted as the terminal moraine of Hansbreen (Ćwiąkała et al., 2018). A series of smaller transverse ridges, plough marks, depression areas, and pockmarks characterize the



bathymetry of inner Hansbukta approaching the terminus of Hansbreen. Hansbreen is a polythermal glacier with a mixed basal thermal regime and covers an area of ~54 km$^2$ (Błaszczyk et al., 2013). The calving front of Hansbreen is 1.7 km long (Błaszczyk et al., 2019), grounding at ~70 m depth, and has retreated on average 38 m yr-1 between 1992-2015 (with a maximum rate of 311 m yr$^{-1}$), which is more than twice its historical rate (Błaszczyk et al., 2021; Grabiec et al., 2018). Svalbard's enhanced warming trends combined with Hansbreen's long history of study (e.g., Kosiba, 1960) and close

proximity to the Polish Polar station (https://hornsund.igf.edu.pl/) makes this an excellent field site for acoustic data collection and methods.



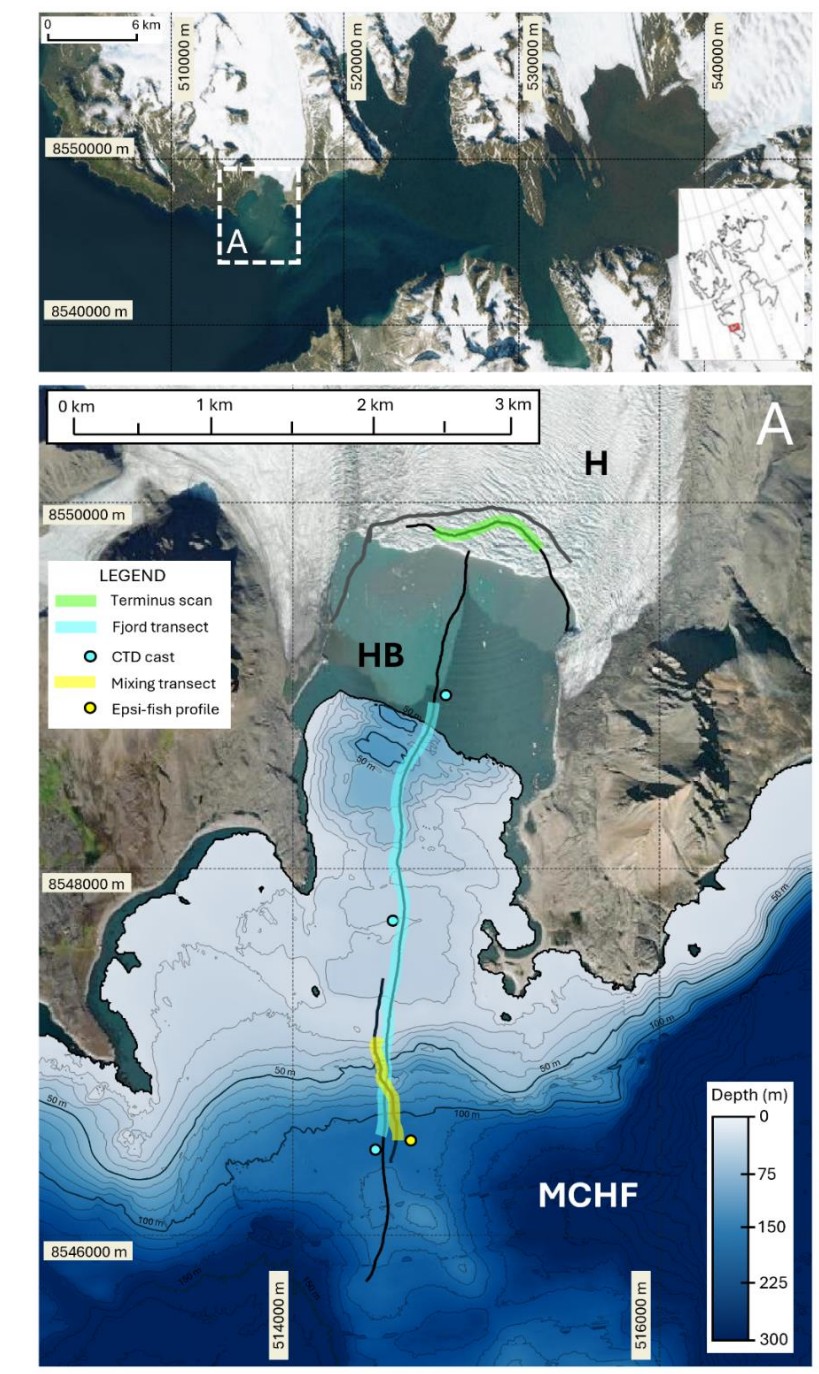

**Figure 1: The general location of 2023 field program in Hornsund Fjord, Spitsbergen, Svalbard is identified in the upper image by the inset box A. The region of study in Hornsund fjord is illustrated in the lower image. The background image is from Landsat−8 imagery collected in June 2023, courtesy of the U.S. Geological Survey, Department of the Interior. Multibeam bathymetry of Hornsund fjord (MCHF) and Hansbugta (HB) is provided by Błaszczyk et al. (2020) and the Norwegian Hydrographic Service.**





**Bathymetric coverage of Hansbugta does not fully cover the region of study due to the retreat of Hansbreen (H) since the most recent seafloor survey. The position of Hansbreen's terminus during operations in July 2023 was estimated using MV Ulla Rinman's radar**
**and is denoted by the grey line. The black lines running north-south across Hansbugta and parallel to Hansbreen are the tracks of MV Ulla Rinman during the broadband acoustic data collection. Highlighted regions identify the extent of echograms from Fig. 3 (blue), Fig. 4 (green), and Fig. 5 (yellow). Circular markers indicate the positions of the CTD casts and Epsi-fish profile used in the analysis of Fig. 3 and 5, respectively.**

## 3 Data collection methods

Data were collected onboard the Marine Vessel (MV) Ulla Rinman and included broadband acoustic water column data, conductivity-temperature-depth (CTD) profiles, microstructure casts, and time-lapse imagery (Fig. 2). Broadband acoustic water column data were collected with a Simrad EK80 wideband transceiver (WBT) transmitting through a Simrad ES200-CD split-beam transducer with a 7° circular beam and frequency range of 160-240 kHz. The transducer was deployed from a 6-meter-long side-mount pole with two acoustic geometries during the field campaign: downward- and side-looking. The

transducer was mounted directly to the vertical pole for the downward looking geometry. For the side-looking geometry, a 3D printed plate adaptor provided a near-horizontal incidence, with a 10° declination angle.

The system was operated continuously during survey operations in broadband mode and all acoustic system parameters (i.e. transmit power, signal mode, pulse length) were kept constant during acquisition. The acoustic system was calibrated using a 25.0-mm tungsten-carbine sphere during field operations on 6 July 2023 and in the Chase Engineering Tank at the University

of New Hampshire with both a 25.0-mm and 38.1-mm tungsten-carbine sphere on 13 March 2024, following the well-documented procedure described in Demer et al. (2006).

Position and attitude data were collected and applied to the field data in post processing from a SBG system Ellipse-D Inertial Navigation System (INS). The INS was mobilized on the bridge and the antennas were mobilized on the bridge deck railing of MV Ulla Rinman, with a clear sky view and more than 2-meters of separation. The INS provided horizontal and vertical

positioning accuracies of 1.2 m and 1.5 m respectively. Pitch and roll accuracies were 0.1° and heading accuracy was 0.2° for 1 m baseline (http:// www.sbg-systems.com).

Ground truth data was collected by two in-situ sensors, a Valeport miniCTD sensor and an Epsilometer microstructure probe, "Epsi-fish" (Le Boyer et al 2021). The Valeport miniCTD unit provided vertical profiles of water column temperature and salinity and was deployed via MV Ulla Rinman's winch. The accuracies of the pressure, conductivity, and temperature

measurements from the Valeport miniCTD unit were ±0.05% of full range (300 m), ±0.01 mS/m, and ±0.01°C, respectively (https://www.valeport.co.uk/). The Epsi-fish, deployed from a commercial fishing reel, provided a total of seven profiles of thermal ($\chi\_T$) and turbulent kinetic energy ($\epsilon$) dissipation rates. The Epsi-fish, which provided measurements of thermal and kinetic energy dissipation rates, was developed at Scripps Institution of Oceanography by the Multiscale Ocean Dynamics (MOD) group (https://www.mod.ucsd.edu/). A full description of the Epsi-fish sensors, accuracies, and deployment

considerations can be found in LeBoyer et al., (2020). Additionally, two GoPro Hero11 time lapse cameras were deployed to capture images of ice coverage (ice mélange and icebergs) and other surface expressions in the fjord during survey operations.



A complete description of the field deployment, data collection procedure and data processing details can be found in Appendix 6.1-6.2.

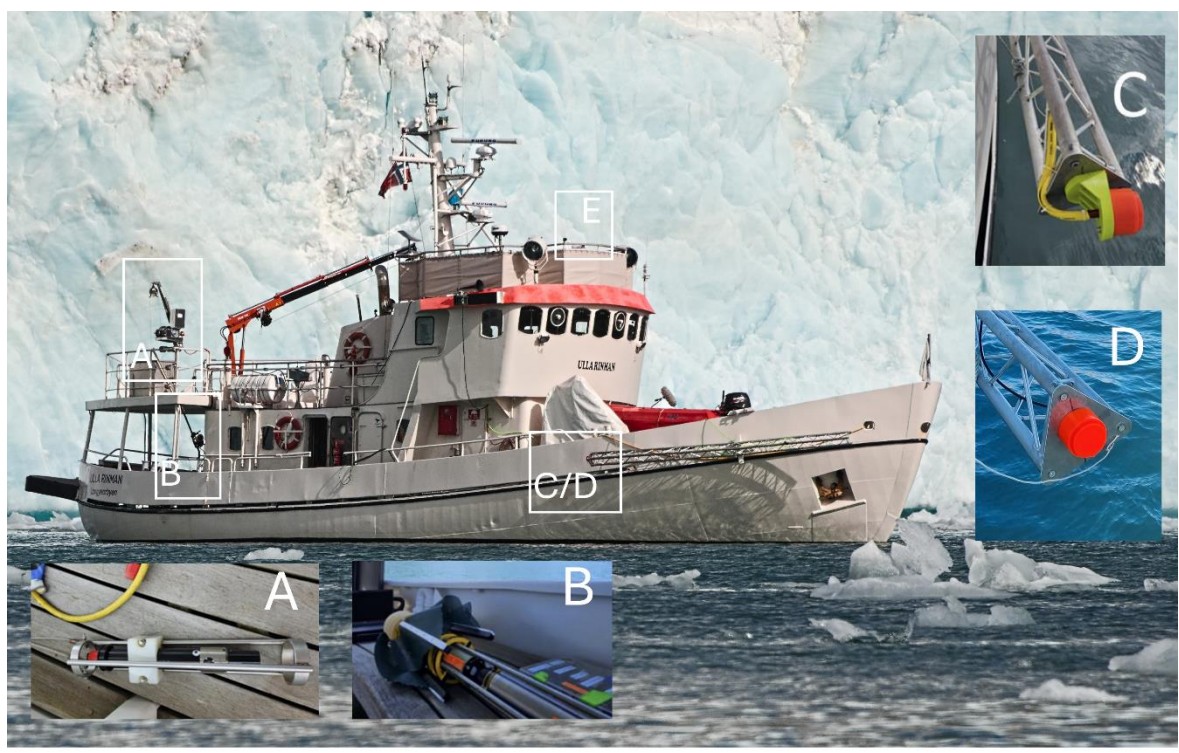

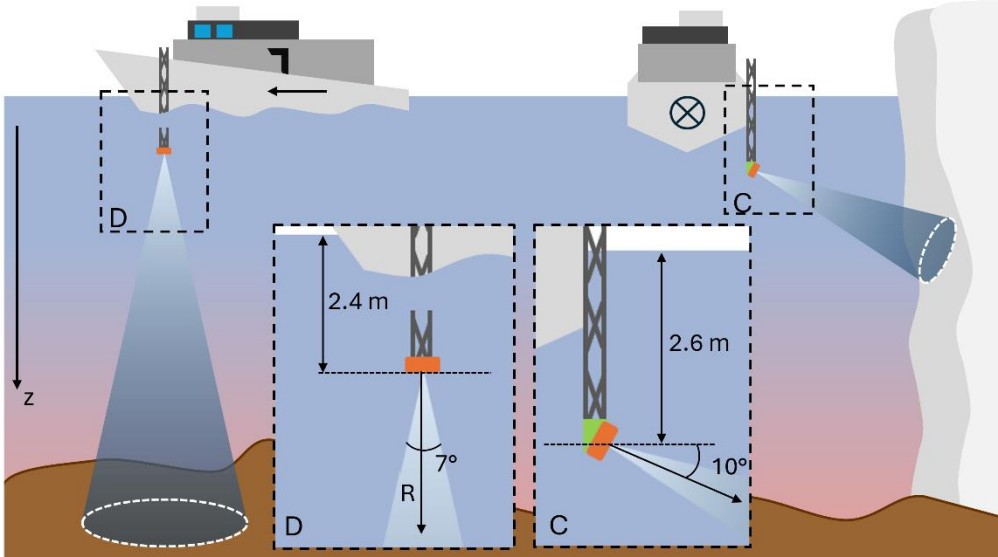


**Figure 2: The acoustic survey platform, MV Ulla Rinman. The ground truth systems, CTD and microstructure profiler, were deployed from the starboard aft of the ship. The CTD sensor was deployed from the ship's winch, inset box A, on the second level,**



and the microstructure probe was deployed from a commerical fishing reel, inset box B, from the main deck. The broadband acoustic transducer was deployed forward, on the starboard side by a side-mounted pole in two geometries, inset box C/D. The side-looking geometry, 10° declination angle from horizontal, was achived with a 3-D printed mount plate adaptor (inset box C) was used to collect acoustic scans of the submerged ice face of Hansbreen. The downward-looking geometry (inset box D) was used to collect fjord-scale transects.

## 4 Interpretation and analysis of broadband echosounder data

This section describes the interpretation and analysis of quantitative acoustic imagery collected by high frequency, broadband, split-beam echosounders. In the underwater acoustic community, experience and familiarity with acoustic data interpretation methods is assumed, and therefore embedded in the literature. However, the aim of this manuscript is to expand the use of broadband acoustic tools in coastal polar regions by engaging a broader audience, so here we cover these topics in detail.

High resolution, near-synoptic images collected by active acoustic systems are commonly referred to as echograms. Echograms consist of a series of individual time series records of acoustic scattering intensity, typically collected at uniform time intervals. In the case of split-beam echosounders, individual acoustic profiles are measurements of backscattering intensity due to the monostatic geometry of the co-located transmit and receive array (i.e., the transducer, see Medwin and Clay (1997), Sect. 10.1). When a split-beam transducer is mounted on a vessel and data is collected underway, as in this work, the sequential stacking of closely spaced acoustic profiles effectively provides an "image" in space and time of the water column (see Fig. 3). In this manuscript, we refer to data collected in this manner as the near-synoptic observations. We use the term near-synoptic because the time interval between individual acoustic records results in an along-track dimension that is always convolved with time, rather than a truly synoptic "snapshot" of the ocean. As a result, observations of non-stationary phenomena (e.g., internal waves, fish schools) can be aliased and this effect should be considered in analysis. Regardless of this complexity, echograms provide highly resolved images of the water column that allow for contextualization and interpretation of oceanic phenomena with resolutions unmatched by other observational means.

There are two types of resolution in an echogram: the along-ray path resolution and along-track resolution. The along-ray path, or range, resolution of an individual acoustic profile is defined by the distance between independent estimates of the backscatter strength. With the application of pulse compression processing techniques, the range resolution of broadband systems is proportional to the inverse of the pulse bandwidth (Chu and Stanton, 1998; Turin, 1960) and typically ranges from millimeter to decimeter scales. The spatial resolution, defined by the distance between individual acoustic profiles in the along-track direction of vessel motion, is a function of the transducer fire rate (typically 0.5-5 Hz) and duty cycle, the depth of the seafloor, and the speed of the vessel. In nearshore environments, when seafloor depths are relatively shallow (<150-m), typical spacing between profiles ranges from sub-meter to meter-scale. Fig. 3 and other data from the Hornsund field site in this manuscript have range resolution of approximately 1.5 mm and spatial resolution between 30 cm to 2 m.

Echograms offer more than just highly resolved images; they contain quantitative information on ocean water column processes and boundaries. Acoustic energy from the outgoing broadband pulse is scattered and reflected from phenomena in both the water column (e.g., fish, turbulent microstructure, suspended sediment) and at boundaries (e.g., seafloor, sea surface,





ice face) that create changes in medium density and sound speed, referred to in the acoustic literature as impedance contrasts. In the context of the ocean, impedance contrasts are often the result of physical boundaries (see Fig. 2 in Weidner et al., 2020), suspended particulates (plankton, fish, sediment, see Fig. 2 in Cotter et al., 2021 or Fig. 2 in Loranger et al., 2022), changes in

thermohaline structure associated pycnoclines (see Fig. 2 in Stranne et al., 2017), and flow or mixing structures (see Fig. 1 in Lavery et al., 2013) within the interior of the water column. The characteristics of the backscattered signal (e.g., intensity, broadband spectral content) are linked to geophysical properties of oceanic phenomenon, which can be characterized through acoustic data analysis. Broadband echosounder data collected in Hornsund fjord is here used to demonstrate methods for backscatter data interpretation and analysis in the following sections: the contextualization and characterization of fjord-scale

thermohaline structure using acoustic observations and direct sampling (Sect. 4.1), leveraging the remote nature of acoustic measurements to observe the largely inaccessible submerged terminus (Sect. 4.2), and the quantitative analysis of tidally and bathymetrically forced mixing through broadband acoustic inversion methods (Sect. 4.3).

## 4.1 Characterizing thermohaline structure in Hansbukta

Characterizing the distribution of water masses in high latitude fjords is complicated by the presence of ocean and glacially

derived water masses (e.g., submarine melting, subglacial discharge), as well as fjord-scale processes with intense spatial and temporal variability (Hager et al., 2022; Straneo et al., 2011). Elevated levels of vertical mixing in fjords are driven by strong, yet intermittent turbulence associated with the tidal cycle and steep bathymetric features, such as terminal moraines (Farmer and Armi, 1999; Schaffer et al., 2020; Smyth and Moum, 2001). Additionally, the degree of stratification (Geyer and Ralston, 2011), local meteorological conditions (Slater et al., 2018b; Straneo and Cenedese, 2015), and sill-driven reflux (Hager et al.,

2022) can influence circulation dynamics and therefore, thermohaline structure.

Traditionally, water masses are characterized through the in-situ collection of conductivity-temperature-depth (CTD) profiles. Thermohaline structure can be inferred by interpolating between CTD profiles - the denser the profile collection, the more accurate the measurements of the spatial distribution of thermohaline structure. Despite their ubiquity and utility, disentangling interconnected processes controlling thermohaline structure can be challenging using direct observational measurements alone.

Individual profiles lack spatial context and dense profile collection can be cost- and time-prohibitive. Acoustic observations can be leveraged to assist in mapping out and characterizing water column structure through imaging of the boundaries (pycnoclines) between water masses. Scattering at water mass boundaries (e.g., pycnoclines) is well documented in the literature and connected to a number of scattering mechanisms, including the sharp gradients in thermohaline structure (e.g., Stranne et al., 2017; Weidner et al., 2020), the intense turbulent mixing due to shear forces (Geyer et al., 2017; Lavery et al.,

2013), and the suspended particles, such as biological aggregations (e.g., Benoit-Bird et al., 2009) or sediment (e.g., Simmons et al., 2020), that are often associated with thermohaline structure. The inclusion of broadband acoustic data together with direct sampling can elucidate the fjord-scale dynamics that control the distribution of thermohaline structure.

During the Hornsund field campaign acoustic transects of Hansbukta were collected alongside in-situ sampling with the miniCTD sensor. The acoustic transects ran along a north-south heading from the deep channel at the mouth of Hornsund





Fjord into Hansbukta to the submerged terminus of Hansbreen (see Fig. 1). The echogram in Fig. 3 illustrates a section of one such transect, approximately 1.7 km long and collected over a period of 35 minutes during the peak ebb tide. The complex, glacially derived bathymetry of Hansbukta is captured in the acoustic record; running from south to north, the bathymetry transitions from the deep (>200-m) main channel of Hornsund fjord to the glacial trough of Hansbreen, where the grounding line is approximately 65-m deep. Between the main channel and Hansbreen the seafloor shallows rapidly over a series of sills,

transitioning rapidly from >200-m to 12-m in depth over less than 250-m along track. Along the echosounder transect, three miniCTD profiles were collected at key locations: in the main channel, at the shallowest point of the primary sill, and in the inner fjord. The CTD profiles provide direct measurements of the thermohaline structure at discrete locations across Hansbukta, while the broadband echogram provides spatial context on water mass evolution and the hydrodynamic processes influencing thermohaline structure.






**Figure 3: An example broadband acoustic transect collected in Hansbukta over the shallow sills separating Hansbreen (far right) from the primary channel of Hornsund fjord (far left). See Figure 1 (blue highlighted transect lines and blue markers) for transect and CTD cast locations within the larger study site. The echogram is colored in a logarithmic scale by volumetric scattering strength (Sv) Three CTD casts were collected, at the beginning (CTD 1) and end of the transect (CTD 3), as well as at the shallow point of the terminal moraine of Hansbreen (CTD 2). CTD profiles of conservative temperature (Θ) and absolute salinity (SA) are plotted against depth in their approximate cast position with reference to the acoustic transect. During CTD cast collection the acoustic data collection was paused, resulting in the break in the echogram over the sill. The maximum depth of the echogram was limited to 45-m to best illustrate the area of interest in thermohaline structure. Regions of particular interest are highlighted in boxes A-D. Temperature-salinity (T-S) diagrams illustrate the connectivity of waters across Hansbukta and the influence of specific water masses. T-S diagrams plot absolute salinity against conservative temperature for all three CTD casts, colored by depth. On each plot the T-S data from the other two casts is plotted in light gray for comparative purposes. Specific water masses are noted, including Transformed Atlantic Water (TAW), region influenced by sill mixing (SM), and the surface layer (SL.**

Generally, thermohaline structure in Hansbukta is influenced by two external water masses: Atlantic water (AW) transported on the West Spitsbergen Current, characterized as both warm and saline (3.5 to 6.0°C, >35 PSU) (Strzelewicz et al., 2022; Walczowski, 2013), and the relatively cold and saline waters (-1.5 to 1.5°C, 34.3 to 34.8 PSU) originating from western Svalbard carried on the Sorkapp Current (SC) (Cottier et al., 2005; Promińska et al., 2018). Additionally, submarine glacial discharge (SGD) and submarine meltwater (SMW) from ablation, melting and runoff associated with Hansbreen are introduced into the glacial trough of Hansbukta. Discharge rates from Hansbreen are variable depending on the season, primarily peaking between June and September (Jakacki et al., 2017).

The three CTD casts can fingerprint the external and glacially influenced water masses and provide a coarse view of the distribution of thermohaline structure across Hansbukta. At the southernmost extent of the transect in the main channel of Hornsund Fjord (>20 m) the CTD profile shows that the deep water column can be described as transformed Atlantic water (TAW – Fig. 3), the product of entrainments and mixing between AW and SC waters (Nilsen et al., 2008); however, the cold and saline TAW waters are isolated by the shallow sill of Hansbukta and are absent in the sill and glacial trough CTD profiles. At the surface, all the CTD profiles show a well-mixed surface layer (SL – Fig. 3), characterized by relatively warm and fresh water, likely influenced by SGD and SMW from Hansbreen. The vertical extent of SL water deepens and warms between the glacial trough and the sills, from 2.0 to 7.5 meters in depth and from 3.6°C to 4.7°C, and then retains the same depth and T-S characteristics between the sill and main channel. Below the surface layer, the thermohaline structure of the water column from the main channel of Hornsund fjord, over the sills, to the glacial trough shows nearly identical T-S characteristics, influenced by sill mixing (SM – Fig. 3). SM water is defined by increasing salinity and a temperature inversion with increasing depth. SM water has a minimum temperature of 2.7°C and shallows in depth from 12.2 to 6.5 meters between the glacial trough and the sill CTD casts. It extends over the water column of the sills and glacial trough CTD casts from the base of the SGD-influenced surface layer to the seafloor, while in the main channel is bounded by a depth approximately equal to that of the shallowest sill depth (~12-m). Overall, the CTD data suggests the influence of Hansbreen extends across Hansbukta well into the main channel of Hornsund fjord evidenced by the persistent, low-salinity SL water across all casts. Furthermore, there is evidence of strong connectivity between the deeper SM water mass in the glacial trough of Hansbukta and in the main channel of Hornsund fjord; SM water is characterized by the low temperature intrusion layer seen in all three T-S plots. SM water is constrained by the sill depth in the main channel, suggesting a bathymetric control on not only the thermohaline structure but




the transport of heat and salt between the main channel of Hornsund and the inner bay of Hansbukta and the ice-ocean interface of Hansbreen. However, without additional casts or other information, disentangling the interconnected processes that are controlling the spatial distribution of SL, SM, and TAW water in Hansbukta and the broader transport of heat and salt across Hornsund fjord (e.g., tides, bathymetry, glacial discharge rates) remains out of reach.

The broadband acoustic data collected between the CTD casts offers the potential to more fully characterize the distribution

of thermohaline structure across Hansbukta and start to explain the dynamics behind evolution observed in the in-situ profiles. The water mass boundaries identified by the CTD casts are clearly visible in the echogram as regions of elevated scattering, as illustrated by the gray dashed lines in Fig. 3. The mechanism in the water column responsible for the elevated scattering signal is not immediately clear from visual inspection of the echogram and could be tied to a number of processes (e.g., gradients in T-S, mixing, biological or sediment particulates); however, these regions show general agreement in their position

(depth) with the in-situ data on water mass boundaries. Unlike the in-situ data, the echogram allows for precise positioning of the visible boundaries across the entire transect of Hansbukta and the nature of the scattered signal (e.g., shape, intensity) provides context on the active mixing processes occurring in the water column.

Starting with the surface layer, SL waters can be identified by the scattering associated with the sharp gradient in temperature and salinity that occurs at the boundary between the base of the surface and the intermediate water mass below. While this

boundary is not visible in the glacial trough (<200 m along track) because the depth of the surface layer is shallower than the draft of the transducer, beyond this point the boundary can be tracked as it progressively deepens from the glacial trough over the sills and to the main channel. The nature of the scattering associated with this boundary evolves along track, elucidating the processes that modify the water mass, as measured in the CTD casts. At the onset of the sills, the boundary is characterized by a series of Kelvin-Helmholtz (KH) instabilities (Fig. 3, inset box A), recognized by their alternating braid-core structure

(e.g., Thorpe, 1987; van Haren and Gostiaux, 2010). KH instabilities are regions of intense shear and turbulent mixing and have been imaged acoustically in other highly energetic coastal regions (Geyer et al., 2017; Lavery et al., 2013). Here, the shear is likely generated by the south flowing current at the surface driven by the ebb tide, potentially combined with the shallowing bathymetry of the sills, evidenced by the presence and directionality of the KH shear instability structure. The mixing between the surface layer and the intermediate water mass below explains the deepening and change in surface layer

properties observed in the CTD casts. Beyond the sills, starting at 1000 m along track (Fig. 3, inset box B), the boundary of the surface layer becomes a discrete, thin scattering layer. The lack of KH instabilities, suggests scattering from a sharp gradient associated with the pycnocline (e.g., Stranne et al., 2017; Weidner et al., 2020), with little mixing or modification in the boundary properties; these observations suggest the surface layer should have similar T-S properties between the sill and main channel, as was observed in the CTD profiles.

Deeper in the water column there is intense scattering from KH shear instabilities closely associated with the rapid changes in bathymetry at the onset (Fig. 3, inset box C) and the end of the sills (Fig. 3, inset box D). In the inner fjord at the onset of each sill, trains of KH instabilities are visible, closely tracking the seafloor depth. The strongest scattering is associated with the second sill, starting at approximately 300-m along track, highlighted in inset box C. Given the close contact with the seafloor,



the elevated scattering levels associated with the KH instabilities could be driven by resuspended sediment from strong
currents, entrained gas bubbles or organisms, or from turbulent microstructure; in either case, the observations from the
echogram points to intense mixing in the inner fjord near the seafloor and transport of water across the sills, likely driven
primarily by the changing tide. On the far side of the outermost sill, entering the main channel of Hornsund fjord, there is a
region of elevated scattering centered at approximately 10-m depth that starts at the sill edge and continuing for an along track
distance of more than 400-m to the end of the echogram record (Fig. 3, inset box D). The depth of elevated scattering matches
the base of the intermediate water mass identified in the CTD casts by the strong temperature inversion, while the nature of
the scattered signal points to an active mixing process although the braided structure of KH instabilities is not as clear as those
in the inner sill area. Again, observations from the echogram point to water mass transport over the sill and explain the
connection in T-S properties between the intermediate waters of the main channel of Hornsund fjord and the inner bay of
Hansbukta in the glacial trough.

Overall, the combination of the near-synoptic acoustic observations and the in-situ CTD measurements point to a combination
of tidally induced flow and the shallow sills controlling the intermediate thermohaline structure across Hansbukta, while a
surface layer influenced by SGD is persistent throughout Hansbukta. Tidal pumping is identified as the driving process for
water mass transport and transformation from 1) the presence and directionality of the KH instabilities that develop at the
interface between water masses and the sills as imaged in the echogram, and 2) the resulting connectivity of thermohaline
structure measured directly by the CTD casts. This type of stratified flow over topography is well defined in the literature (e.g.,
Farmer and Armi, 1999; Geyer et al., 2017) and has been observed in estuaries, as well as in terrestrial systems such as
mountain ranges (Lilly, 1978). This conclusion is supported by previous research in Hornsund fjord, which suggests that tidal
forcing is the main hydrodynamic driver of heat, salt and fresh water budgets (Jakacki et al., 2017; Kowalik et al., 2015). The
analysis of Fig. 3 transect illustrates the utility of broadband echosounder data as a contextual tool for the study of high latitude
systems and the importance of direct sampling sensors, such as CTDs to inform the analysis and vice versa. The CTD profiles
lack the high-resolution spatial context to directly determine the dynamics driving the thermohaline variability, while the
acoustic observations are inherently ambiguous, requiring ground truth information for a complete analysis. Here, a
combination of the near-synoptic acoustic observations with the in-situ measurements of the thermohaline structure can
provide important physical context for interpretation of the driving processes in high latitude systems.

## 4.2 Remote observations of the ice-ocean interface

Beyond the near-synoptic observations, echosounders can collect measurements in regions inaccessible to direct sampling
equipment. In high latitude glacial fjords this remote advantage can be leveraged to collect measurements in dangerous regions,
such as the submerged ice-ocean interface of the glacial terminus. Direct observations at the ice edge in literature are few
owing to limited access by the presence of ice mélange and ice bergs. Furthermore, calving events, both subaerial and
submarine, make work at the ice face dangerous for vessels, equipment, and personnel, and preclude most ship-based sampling.
Oceanographic data, such as CTD casts, are generally collected downstream in the fjord and ice-ocean dynamics are then



inferred (Beaird et al., 2023; Stevens et al., 2016; Straneo et al., 2011). The few in-situ observations have been collected by helicopters (Bendtsen et al., 2015), remotely operated vehicles (Mankoff et al., 2016), and sensors attached to marine mammals (Everett et al., 2018). The paucity of direct measurements limits the inclusion of terminus ice-ocean dynamics in large-scale

climate models (Bamber et al., 2018), predictions of fjord circulation (Straneo et al., 2011), and contribution to sea level rise predictions (Siegert et al., 2020). The need for enhanced observational capabilities and additional measurements at the ice-ocean interface has been well documented, e.g. Siegert et al. (2020).

A number of recent studies have employed acoustic systems to overcome the challenges associated with near-terminus observations. Sutherland et al., (2019) estimated submarine melt rates by combining high resolution images of the submerged

terminus from a multibeam echosounder with data on glacier motion. An acoustic Doppler current profiler (ADCP) deployed from an autonomous surface vehicle by Jackson et al. (2020), provided the first characterization of meltwater intrusions in the near-terminus region. Broadband echosounders can add to this acoustic toolset by providing remote, quantitative observations of the ice-ocean interface and those processes occurring nearby, such as subglacial discharge. The possible data collection proximity to the terminus is constrained by the range dependent signal to noise ratio (SNR), which is strongly frequency

dependent, owing to the increasing impact of molecular relaxation of magnesium sulphate molecules with increasing frequency (Lurton, 2002). The broadband ES200-7CD used in this work has a range threshold of approximately 200 meters before SNR is insufficient for data analysis. Within this range, broadband echosounders uniquely can make measurements of absolute scattering intensity, not possible with other acoustic systems, and these measurements, combined with split-aperture processing and broadband spectral analysis, can shed light on the morphology of the terminus, identify the presence and constrain the

geometry of subglacial discharge plumes, and identify inner-connection bio-physical processes occurring near the terminus.

During the 2023 field campaign in Hansbukta, measurements of the submerged terminus of Hans Glacier were collected using a side-looking geometry (Fig. 2). A series of survey lines were run for approximately 1.2 km parallel to the ice face from east to west, limited by the shallowing seafloor edges of Hansbukta. The transducer geometry provided a diagonal slice of water column observations starting at 2.6 m depth (transducer draft) to between 17-25 meters deep at the submerged terminus (see

Appendix 6.2 for additional information). The resulting echogram provides imagery of the submerged terminus, as well as near-terminus water column phenomena occurring between the transducer and the ice face, including scattering from subglacial discharge (Fig. 4). Concurrently with broadband data collection, water surface imagery was collected with a time-lapse camera. During survey operations near Hansbreen the surface expression of a subglacial discharge plume was observed as a persistent ice-free region extending outwards from the ice face <200 m. No surface flow velocity measurements were made but a distinct

boundary was observed at the edge of the plume, marked by the ring of ice mélange and a change in surface texture. Observations are consistent with the surfacing of buoyant subglacial discharge reported in the literature (Chauché et al., 2014; Mankoff et al., 2016; Mortensen et al., 2013; Motyka et al., 2003; Xu et al., 2013)





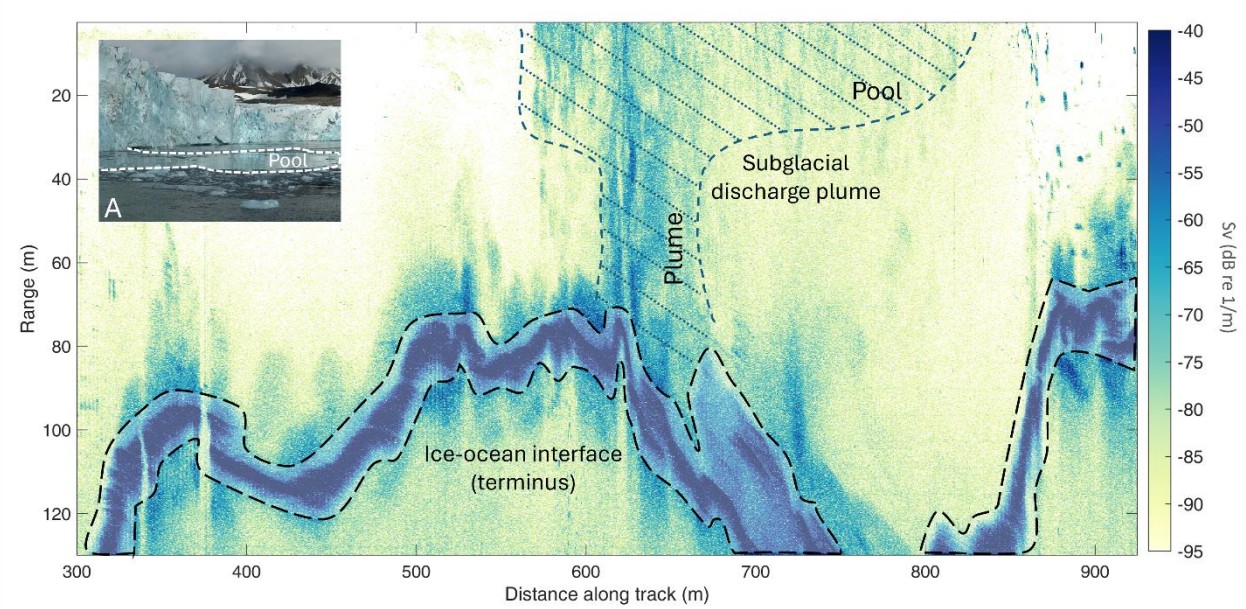

**Figure 4: Broadband acoustic observations of the submerged terminus of Hansbreen and the surface expression from the time-lapse imagery (inset box A). See Fig. 1 (green highlighted transect lines) for transect location within the larger study site. The ice-ocean interface of Hansbreen is bounded by the dashed black line. The submerged expression of the subglacial discharge plume and its evolution to the surface pool is marked by the dark blue dashed line.**

Broadband echosounders can precisely position the submerged terminus, as well as provide measurements of the ice face along-track geometry (Fig. 4). In the echogram the terminus is easily identifiable across the entire transect by an elongated, strong backscattering return starting between 75 to 150 meters in range. The elongated return from the submerged ice face is initially characterized by the side lobe return, a weakly scattering signal of a variable length. The primary return from the submerged terminus shows a generally monotonic increase to a peak intensity value of between -45 to -35 dB re 1 µPa, before a gradual reduction in intensity. The length of the primary return is variable, likely due to variability in the angle of ice face tilt with respect to the transducer. This type of elongated return resulting from oblique incidence geometry and the propagation of the acoustic pulse along an extended surface target has been observed in other broadband echosounder work (e.g., Weber and Ward, 2015), as well as in multibeam echosounders at large incidence angle (e.g., Lurton, 2002 Fig. 8.30). Tracking the terminus position can provide a direct measurement of submerged terminus geometry. Future efforts with such a data set could provide:

1)  General trends in the over- or -under-cutting of the submerged ice face with respect to the above-water terminus (e.g., Abib et al., 2023; Rignot et al., 2010; Robertson et al., 2012; Sugiyama et al., 2019),

2)  measurements of terminus morphology over the scanned region,





3)   the means to connect terminus-adjacent processes (e.g., subglacial discharge plumes, calving events, bubble injection)
to variability in geometry, and

4)   measurements of localized submarine melt rates across the length of terminus from repeat transects combined with
         glacier motion estimates, similar to Sutherland et al., (2019).

In addition to observations of the submerged terminus, elevated backscattering from subglacial discharge can be seen in the echogram between 575 and 825 meters along the transect. The geometry of the subglacial discharge evolves as it rises in the

water column towards the surface. Initially, the region of elevated scattering extends outwards from the submerged terminus in a narrow core approximately 50 meters across (620 to 675 meters along track), referred to here as the "plume" region following Mankoff et al. (2016). This plume extends outward from the ice face 40 to 50 meters with little change in its along-track width (parallel to the ice face). At a range from the transducer of approximately 25 meters, 7 meters depth, the plume rapidly expands to a region more than 300 meters across, referred to here as the "pool". The surface expression of the subglacial

discharge pool, observed from the deck of MV Ulla Rinman and in the time-lapse imagery, generally agreed with the acoustic measurement of pool width. To our knowledge, these are the most detailed measurements, acoustic or otherwise, of the geometry of subglacial discharge event. Measurements such as these can provide direct information about the variability in plume width, which strongly affects the plume neutral buoyancy depth, upwelling flux, and plume-driven renewal time (Slater et al., 2022). Future analysis could take advantage of split-aperture processing techniques employed for gas seep plumes (e.g.,

Blomberg et al., 2018; Jerram et al., 2015) to determine the position and geometry of the plume and submerged ice face more precisely.

Beyond spatial measurements of the discharge, broadband acoustic observations can elucidate the dynamics of subglacial discharge plumes, such as the scale and geometry of the discharge location at the grounding line. It is well established that fresh water from surface melting, precipitation events, and other terrestrial inputs is channeled upstream of the terminus and

is injected into the fjord environment at the calving front; however, the nature of the discharge location, whether at single or multiple distinct locations (point source) or across the length of the terminus (line source) remains unanswered (Jenkins, 2011; Mugford and Dowdeswell, 2011; Sciascia et al., 2013; Xu et al., 2013, 2012). The geometry of the discharge location defines the plume evolution and subsequently the entrainment rate of ambient fjord water, the total area of enhanced submarine melt rates due plume-ice interaction, and the spread of glacially modified waters at the depth of hydrostatic equilibrium (Motyka et

al., 2003; Wagner et al., 2019). Moreover, geometry is an important component of the parameterization of plume dynamics for fjord-scale modeling (Cowton et al., 2015; Slater et al., 2020). With the proper deployment geometry, the broadband system can provide direct observations of the subglacial discharge location; however, even without direct observation, the backscattering signal from the plume higher in the water column can provide important observations related to plume dynamics.

Acoustic observations from Hansbukta demonstrate echosounders can verify the presence of subglacial discharge in the water column. In Hansbukta the submerged expression of discharge coincided with a surface expression. Prior to this work, the detection of presence of subglacial discharge plume depended on visible surface expression of sediment laden waters (e.g.,



Chauché et al., 2014); however, at termini with deeper grounding lines surface expression is not always expected (e.g., Stevens et al., 2016). Recent modeling work from Slater et al. (2022) indicates that only 28% of subglacial discharge plumes in

Greenland reach a neutral buoyancy in the photic zone due to a combination of near-terminus stratification and weak discharge. The Slater et al. (2022) results suggest that more than two-thirds of cases plume dynamics are under accounted for in modeling efforts of ice-ocean interactions, including increased submarine melt and calving rates; this does not even account for the fact that a plume reaching the photic zone does not guarantee that it will have a visible surface expression. Again, broadband echosounders offer the means to close these observational gaps by providing remote verification of plume presence, as well as

help better understand the dependency of plume dynamics on the slope of the submerged terminus (Jenkins, 2011), identify the plume neutral buoyancy depth, and impact on downstream thermohaline structure, and plume and terminus geometry.

## 4.3 Quantitative analysis and geophysical parameter estimation

Beyond providing near-synoptic observations of the environment, quantitative analysis of broadband spectral data can be used to identify scattering mechanisms and even provide remote measurements of geophysical signals. These methods compare the

calibrated, absolute scattering measurements of a specific phenomenon made by field echosounders to predicted scattering levels from theoretical acoustic scattering models. The models are generally informed by in-situ measurements of the physical (e.g., thermohaline structure, bubbles, suspended sediments) or biological scattering mechanism, collected in tandem with the acoustic field measurements. The direct comparison of scattering levels allows for inversion for geophysical parameters from rapid, high resolution, near synoptic, echosounder observations and offers a promising alternative to traditional direct sampling

approaches in challenging locations, such as high latitude fjords.

The broadband acoustic inversion method applied here has previously been leveraged to measure dissipation rates of turbulent kinetic energy in high energy systems with intense salinity gradients in the Connecticut River (160–600 kHz, Lavery et al., 2013). In highly turbulent strongly stratified water columns, backscattering can be driven by small, random perturbations in density and compressibility brought about by the fluctuations of the medium temperature and salinity associated with dynamic

mixing (Goodman and Forbes 1990). A well-known theoretical acoustic scattering model for isotropic homogenous turbulent microstructure (Lavery et al., 2003) has been used to remotely measure dissipation rates using both narrow and broadband acoustic systems (e.g., Ross and Lueck, 2005; Lavery et al., 2013; Muchowski et al., 2022). The scattering model used here, first applied in Lavery et al., 2013, is outlined in detail in Appendix 6.3. In short, the calibrated and corrected volumetric scattering derived from field observations is fit to the scattering model output, which is constrained by the in-situ data collected

by the Epsi-fish, which was <150-m from the acoustic transect. The fit of the scattering model then provides an acoustically derived measurement of geophysical parameters, in this case dissipation rate of turbulent kinetic energy (TKE), $\varepsilon$.

During the Hornsund fjord field program, broadband acoustic observations were collected in tandem with measurements of dissipation rates temperature variance ($\chi_T$) and turbulent kinetic energy ($\varepsilon$) by the Epsi-fish in regions of stratified turbulence (Fig. 5). Comparison between broadband inversion measurements and the Epsi-fish ground truth dataset illustrates the potential

for accurate, remote, acoustically derived measurements of dissipation rates in high latitude regions. Furthermore, the high-





resolution echograms collected by broadband echosounders systems can image and quantify the evolution of shear instabilities associated with mixing over large regions. This is of particular use in highly dynamic, strongly stratified regions such as estuaries or fjords where dissipation rates vary dramatically over the tidal cycle (e.g., Scully et al., 2009; Lavery et al., 2013; Geyer et al., 2010; in these regions direct observations are limited by cost and deployment challenges.

The application of broadband acoustic inversion to remotely measure dissipation rate TKE was applied to the acoustic transect collected running from north to south into the main channel of Hornsund Fjord, the area of interest just outside the primary sill of Hansbukta (Fig. 1). The data were collected near the peak ebb tide and the echogram shows elevated scattering associated with shear instabilities, thermohaline structure, and individual fish in the region of interest highlighted in Fig. 5, between 10- and 28-meters depth. The vertical profiles of temperature and salinity from the coincident Epsi-fish cast show several

pycnocline regions associated with glacially derived water mass, similar to the structure observed in Fig. 3 and discussed in Sect. 4.1. At the sharp thermocline at ~16-m depth, TKE and thermal dissipation rates observed from the Epsi-fish are elevated more than an order of magnitude above background levels. Elevated scattering is seen at the same location, associated with shear instabilities that evolve along track into a breaking internal wave.

   The dissipation rate of TKE can be measured from the backscattered signal with the help of a theoretical acoustic scattering

model and direct sampling from the Epsi-fish. The Epsi-fish measurements not only inform the acoustic scattering model but were used to validate the acoustically derived measurements. The local temperature and salinity measured by the Epsi-fish (dark gray region, Fig. 5) were used to estimate the molecular kinematic viscosity ($v$), the saline contraction and thermal expansion coefficients ($\beta$ and $\alpha$), the fractional change in sound speed due to temperature and salinity ($\underline{a}$ and $b$), and the mean gradients of salinity ($\partial \bar{S}/\partial z$) and temperature ($\partial \bar{T}/\partial z$). Together with peak values of the dissipation rate of TKE ($\varepsilon$), the Epsi-

fish measurements were used to:

     1)   determine the turbulent energy regime of the broadband acoustic observations,

     2)   model the predicted frequency dependent backscattering cross section ($\sigma_v$) and

     3)   validate our acoustically derived estimate of dissipation rate of TKE ($\varepsilon$).





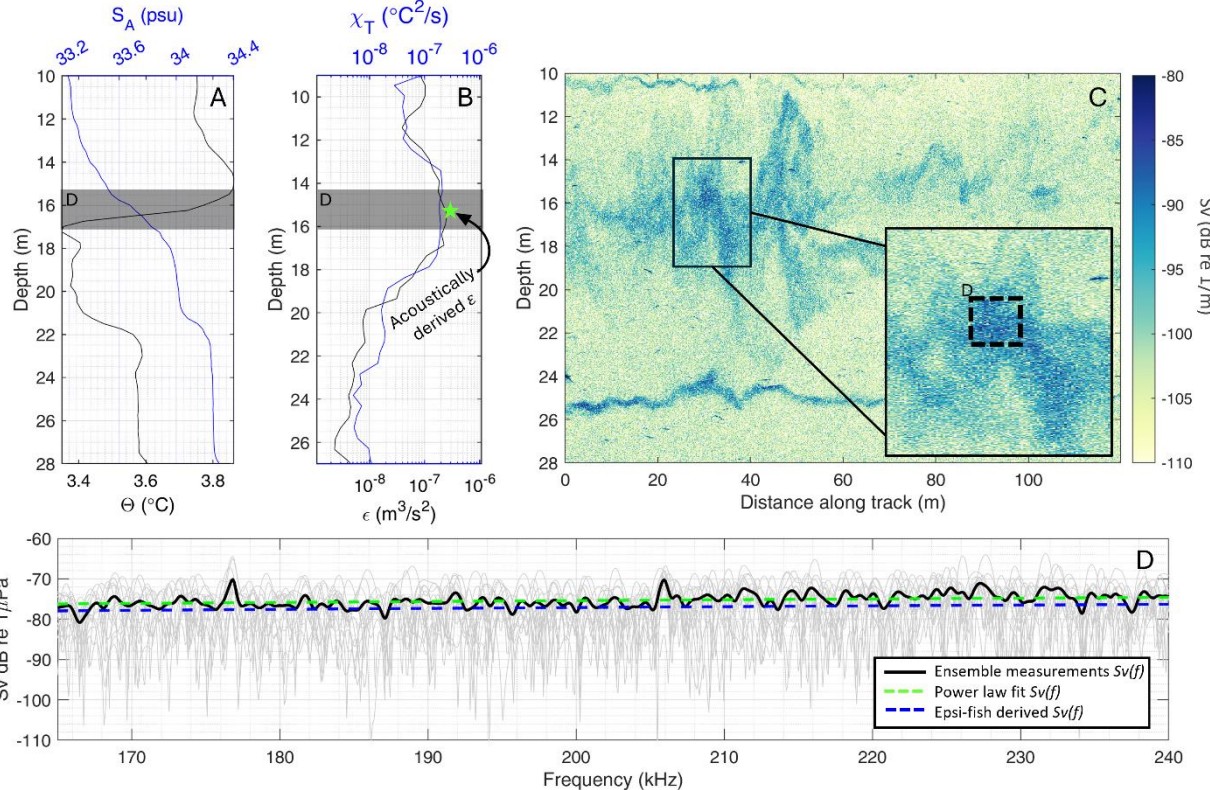

**Figure 5: Combining Epsi-fish ground truth and broadband acoustic data to better characterisze mixing in the water column.** **Vertical profiles of temperature and salinity (panel A), and dissipations rates of temperature variance and turbulent kinetic energy** **(panel B) measured by the Epsi-fish cast. Panel C is broadband echogram of backscattering intensity (Sv) collected in close proximity** **(within 150 m) to the Epsi-fish cast. The inset box in C provides an enlarged view of the sampled region (dashed box marked D),** **from which Sv(f) was calculated using Eq. (2) and is plotted in panel D. The light grey lines in panel D are the individual** **measurements of Sv(f) from each sampled acoustic record and the thick black line is the ensemble average of all records. The dashed** **green line is the power law fit of the ensemble Sv(f) measurement, which was used to invert for ε. The acoustically derived ε is plotted** **in panel B (green start). The dashed blue line in panel D is the output of Eq. (1) evaluated with the Epsi-fish direct measurements,** **taken from the regions highlighted with the dark box in panels A and B. See Fig. 1 (yellow highlighted transect lines and yellow** **marker) for transect and Epsi-fish profile locations within the larger study site of Hansbukta.**

The Epsi-fish measurements of the region of interest in Fig. 5 indicate all the acoustic observations fell well within the viscous-convective turbulent energy regime. As a result, the scattered energy from the turbulent mixing associated with breaking internal wave, here referred to as the backscattering cross section ($\sigma_v$), could be computed following Eq. (6) from Lavery et al. (2013):

$$\sigma_v(f) = 2^{-3} f q \left(\frac{\varepsilon}{v}\right)^{-\frac{1}{2}} \frac{4\pi\varepsilon R_F}{cg\beta} \frac{\partial\bar{S}}{\partial z} \left(A^2 \left(\frac{\partial\bar{T}/dz}{\partial\bar{S}/dz}\right)^2 + B^2 + 2AB \frac{\partial\bar{T}/dz}{\partial\bar{S}/dz}\right) \tag{1}$$

where $f$ is the acoustic frequency in kHz, evaluated over the range 160-240 kHz, $q$ is a constant defined by Oakey (1982), $c$ is the speed of sound, $R_f$ is the non-dimensional Richardson Flux number, defined by Osborn (1980) as 0.15, $g$ is the





gravitational constant, and $A$ and $B$ account for the fractional changes in sound speed and density due to temperature and
salinity, such that $A=\alpha+a$ and $B=\beta-b$.

Equation (1) was used to predict the broadband backscattering intensity spectra over the experimental frequency range (160-240 kHz) using the Epsi-fish measurement (Fig. 5 – blue line). The Epsi-fish derived predictions of backscattering cross section were then compared to acoustically derived measurements of backscattering cross section in the region of interest.

To compute $\sigma_v(f)$, a time series with a length of 36 samples, approximately 0.28 milliseconds, was extracted across 25 acoustic profiles. The Fourier transform of the profiles was taken to compute the frequency dependent backscattering ($S_{mf}$). The ensemble average, backscattering cross section was computed from the ensemble of the extracted $S_{mf}$ measurements following a modified form from Weber and Ward (2015):

$$\sigma_v(f) = \frac{\langle |S_{mf}(f)|^2 \rangle}{C(f)} \frac{r^4}{exp(-4\alpha_a(f)r)} \frac{1}{V(f)} \tag{2}$$

where the brackets $\langle\rangle$ specify an ensemble average over multiple acoustic profiles, $C$ is the frequency-dependent main response axis (MRA) correction factor, $r$ is the range to the region of interest, $\alpha_a$ is the frequency-dependent absorption coefficient in neper/meter, and $V$ is the frequency dependent ensonified area. The volumetric backscattering intensity, plotted in Fig. 5, is the logarithmic form of backscattering cross section, such that $Sv=10\ log_{10}(\sigma_v)$.

Taking the analysis of the acoustic measurements can be one step further, the measurements of backscattering cross section can be inverted using Eq. (1) to directly, remotely estimate the dissipation rate of TKE. The model (Eq. (1)) was rewritten to solve for the dissipation rate of TKE using both the ensemble average volume scattering measurements and the basic oceanographic measurements from the Epsi-cast, and then evaluated to provide a remote, acoustically derived measurement of the dissipation rate of TKE which can be compared to the Epsi-fish in-situ measurement. This method is not without precedent; it has been successfully applied in the Connecticut River by Lavery et al. (2013) and in the Baltic Sea by Muchowski et al. (2022).

Before discussing the results of the broadband measurements of frequency dependent backscattering cross section and the resulting inversion estimating the dissipation rate of TKE, it is important to clarify the assumptions involved in this analysis. First, there is an approximately 150-m separation between the deployment location of the microstructure probe and the acoustic data shown in Fig. 5. Both data were taken approximately the same distance from the sill, but the measurements made by the Epsi-fish may not be representative of the water column sampled by the acoustic system, especially if mixing associated with the sill had spatial heterogeneity. Furthermore, the acoustic scattering model (Eq. (1)) used in this analysis assumes homogenous, isotropic turbulent microstructure is responsible for the scattered signal and any patchiness in that phenomenon is not accounted for in the model application. The short ranges associated with this analysis mean the area sampled is relatively small, <3-m3, and previous research (e.g., Muchowski et al., 2022), working in deeper conditions (larger sampled areas) have made the homogenous, isotropic assumption. Finally, the use of a single acoustic scattering model for inversion estimates assumes either a single or dominant scattering mechanism. It is always possible, indeed likely, that multiple scattering mechanisms are present in the sampled region at a given moment in time; however, this ambiguity can be broken by considering



the broadband spectral response and comparing field measurements to the known frequency dependencies of various potential scattering mechanisms, as is done in the following paragraph.

Challenges in applying acoustic inversion methods notwithstanding, there is excellent agreement between both the modeled predictions of volumetric scattering and the values of dissipation rate of TKE measured by the Epsi-fish cast and the acoustically derived observations as illustrated in Fig. 5. First, the acoustic field measurements of volume scattering from the region associated with the breaking internal wave show a weakly positive frequency dependence of $f^1$, measured by least-square fitting a power law distribution. This power law dependence is the same as that predicted by Eq. (1) for backscattering from turbulent microstructure in the viscous-convective regime. While there are indications of other scattering mechanisms in the echogram (e.g., fish, thermohaline structure), neither have this power law spectra dependency; this combined with Eq. (1) and the general context suggests that the elevated scattering observed associated with the region centered at 16-meter depth is driven by turbulent microstructure. Here, the dissipation rate of TKE derived through acoustic inversion was measured to be $5.5 \times 10^{-7}$ ($m^3/s^2$); compared to the peak Epsi-fish measurement of $2.7 \times 10^{-7}$ ($m^3/s^2$). The close agreement, within a factor of two, suggests that the acoustic method is viable, and that the backscattering signal observed in the echogram surrounding the region of interest could be inverted to provide estimates of the dissipation rate of TKE over a larger area, leveraging the high-resolution nature of acoustic data to characterize mixing across the region.

The next step of this analysis is to expand the acoustically derived measurements to contextualize dissipation rate variability at the sill over various stages in the tidal cycle and over the subglacial discharge plume in the glacial trough. The highly resolved nature of broadband echograms, defined by both individual acoustic profile depth resolution and the spacing of along-track profiles, means we can extract measurements of dissipation rates at spatial scales unheard of with direct profile sampling equipment and use these high-resolution measurements to better inform mixing models in the highly dynamic high latitude fjords; such information can improve our understanding of heat and salt transport across fjords. The fact that TKE dissipation rates that can be obtained remotely is going to be a critical tool for understanding ablation effects at the glacier terminus, where it is very difficult to obtain in-situ measurements.

## 5 Considerations in the use of broadband active acoustic tools

In this manuscript high frequency, broadband echosounders have been introduced as an innovative tool for the study of high latitude glacial fjord systems. In these isolated, often dangerous environments understanding the complex dynamics between ice and ocean is crucial as the region continues to warm and the gap between current observational capabilities and needs remains a central challenge. Broadband echosounders are particularly effective in high latitude fjords because of the distinct water masses that influence the thermohaline structure of the water column and the intense mixing associated with tides, bathymetry, and glaciers found in these regions. These factors give rise to strong backscattering that can be both visualized in a high-resolution spatial context and quantified in numerous ways, with and without the support of direct sampling systems. Furthermore, many high frequency broadband echosounders are small and portable, with relatively straightforward deployment





geometries and mobilization needs, making them an appealing tool for high latitude field work when time and space is limited

(e.g., Cade et al., 2022). The addition of broadband echosounders to a high latitude field kit could allow for continuous water column monitoring between direct sampling stations without any additional deployment or time required from the field team, highlighting the potential in improved analysis with minimal effort expended. The prospects for future studies of high latitude glacial fjords with these systems are excellent, as they can provide critical observations needed to understand the ongoing climate driven changes, particularly surrounding the glacial terminus where direct sampling is challenging.

Despite their many strengths, broadband acoustic systems are not generally used as a stand-alone observational tool and the successful collection of quality data depends on thorough and well-defined mobilization, deployment, survey procedures, field processing and data QA/QC methods. First, while split-beam transducers have straightforward deployment geometries and minimal positioning and motion data stream requirements in comparison to other types of active acoustic systems (e.g., multibeam echosounders), broadband transceivers are sensitive to external noise sources (e.g., other active acoustic systems,

ship noise, electrical interference). The successful collection of quality broadband water column data requires a well-considered mobilization plan for field activities, discussed in Sect. 5.1 below. Second, quantitative analysis, such as broadband spectral characterization or acoustic inversion procedures, require calibration of the split-beam echosounder and application of a series of corrections related to sound propagation and scattering. Sect. 5.2 covers the needs associated with these corrections and the budget of time and personnel for these activities during field expeditions. Finally, Sect. 5.3 discusses the

inherent ambiguity in acoustic observations briefly noted in Sect. 4 and the ongoing challenge in interpreting all active acoustic data, no matter the quality of data collection.

**5.1 Field deployment noise considerations**

Broadband split-beam echosounders are engineered with high receive sensitivity, which allows for the conversion of a wide frequency range of very low amplitude acoustic signals into electrical signals. High receive sensitivity enables the observation

of weakly scattering phenomenon, such as thermohaline and mixing structure, as reported in Fig. 3-5; however, high receive sensitivity also makes broadband echosounders susceptible to external noise, both from other acoustic systems (e.g., ship bottom pingers), environmental sources (e.g., wind, waves), ship noise, and electrical interference. From a practical perspective, any increase in interference to the received acoustic signal from external noise sources will reduce the SNR of the system, which results in a reduction of the range of detection of objects in the water column and broadband spectral

characterization capabilities.

Vessel mobilization of the broadband transducers must consider equipment placement with respect to noise sources and flow structures, as well as transducer safety in high latitude oceans, to ensure the collection of quality data. During the Hornsund fjord field work the ES200-7CD split-beam transducer was deployed from a side-mounted pole on the forward, starboard side of MV Ulla Rinman. The side-mounted pole was a total of 6 meters in length, with 2-meters sitting below the water line to

625 protect the transducer from interaction with surface ice mélange. The deeper the transducer, generally the greater the reduction in acoustical ambient noise sources like waves, wind, rain or ship traffic. The location of the pole was selected to separate the





transducer from the mechanical noise and bubble wash associated with the ship's propulsion system, which was located at the aft of the ship. Additionally, the pole location on MV Ulla Rinman also allowed the field team to maintain visual contact during survey activities from the ship's bridge, as well as easy access for regular pole deployment and acoustic calibration procedures. Installation of the transducer must ensure as close to a laminar waterflow over the transducer face as possible, as any vibration will be converted into electrical energy, contributing to the background noise level.

Noise reduction measures must also be considered in the installation and acquisition procedures of the acoustic topside unit. When the EK80 wideband transceiver was acquiring acoustic data during the Hornsund fjord expedition the MV Ulla Rinman's ship sonar was turned off to remove potential for crosstalk between the two systems. During collection of in-situ data the broadband system did not acquire data, due to interference from the ship's winch and at the completion of in-situ data collection the winch and all associated electronics were secured. Furthermore, the full suite of broadband echosounder equipment (e.g., transducer, transceiver, positioning system, acquisition laptop) was run from a separate marine-grade battery, due to electrical noise associated with ship power.

## 5.2 Acoustic corrections in post-processing

The successful interpretation of acoustic data relies upon the accurate and thoughtful application of corrections in the post-processing pipeline. Acoustic pressure time series require several corrections before thorough analysis of backscattering intensity and/or broadband spectral analysis can commence. Appropriate corrections are dependent on the type of analysis, but typically include the application of a calibration offset to account for the accuracy and precision in signal transduction and measurements, geometric spreading corrections, and frequency-dependent absorption estimations, see Eq. (2).

Acoustic instruments need to be calibrated frequently, as their performance will change with the deployment environment and over the life of the instrument (Demer and Hewitt, 1993; Brierley et al., 1998a; Nam et al., 2007). It is best practice to calibrate as system over the range of environmental conditions encountered when measurements are made; water temperature is of particular concern for high latitude work. Calibration methods for split-beam echosounders are well defined in the literature and are described in detail in Demer et al. (2015). In short, a small (<50 mm) sphere of known physical properties is hung in the beam of the acoustic transducer on the experiment platform and moved throughout the acoustic beam to measure variability in sensitivity. The expected backscattering from the sphere is calculated using a theoretical model for the reflection of sound by elastic spheres (MacLennan, 1981) and is compared to the measurements made in the field. The resulting calibration offset ($C(f)$ in Eq. (2)) adjusts the gain and filter attenuation correction factor of echosounder measurements, as well as correcting for frequency-dependent beamwidth and receive sensitivity inherent to broadband echosounders (Rogers and Van Buren, 1978).

Beyond calibration offsets, all quantitative acoustic analysis must account for the frequency dependent absorption of acoustic energy by the propagation medium and the geometric spreading of the acoustic signal. Acoustic absorption refers to the gradual loss of energy due to interaction with water molecules and dissolved substances. Absorption is a function of both seawater properties (e.g., temperature, salinity), as well as frequency and is corrected for in logarithmic space for frequencies between



160-240 kHz in dB/m (Francois and Garrison, 1982). Geometric spreading refers to the propagation of the broadband signal outward in all directions from the transducer resulting in a range-dependent loss of acoustic energy for a given area. Depending on the underlying physics of the scattering phenomena in question, geometric spreading can be accounted for with different range dependent correction, the most common being spherical spreading correction, but other corrections may be more appropriate depending on circumstances.

The post-processing pipeline for broadband split-beam data to account for these corrections remains a barrier for new users and applications. Many research groups rely on processing tools developed in-house because of the lack of open-source commercial software. Techniques for processing are improving, several open-source data analysis tools have become available in recent years, such as EPS3 (Ladroit and et al., 2023) and EchoPype (Lee et al., 2021). Commercial developers have tools to process narrowband split-beam data and have recently started to expand processes capability to broadband systems. However,

these software packages are costly and their broadband spectral analysis tools are still in the early stages of development.

## 5.3 Ambiguity in the backscattered signal

Observations made by active acoustic systems are inherently ambiguous, as scattering in the water column can be associated with multiple dynamics and visually differentiating between processes is not always possible. The correct interpretation of backscattering signals requires a deep understanding into both how acoustic systems work and proper application of corrections

for sound propagation and scattering associated with targets, like those discussed in Sect. 5.2. In circumstances where these steps are skipped, the potential for inaccurately identifying water column processes and mis-quantifying geophysical parameters is high.

As discussed in Sect. 4.3, broadband spectral analysis combined with theoretical acoustic scattering models can differentiate between scattering mechanisms; however, direct measurements remain an essential and straightforward tool to inform and

contextualize remote acoustic observations. Field procedures should plan to include co-located in-situ sampling when possible; these data can provide context and clarity to the acoustic observations, as illustrated in Sect. 4.1, as well as inform theorical acoustic scattering model to assist in broadband spectral analysis, as in Sect. 4.3.

The application of acoustic scattering models to explain broadband spectral characteristics or assist in acoustic inversion methods also requires forethought and experience. Many broadband inversion methods have been developed to apply to cases

in which one scattering mechanism dominates the ensonified volume, e.g., sizing gas bubbles (Weidner et al., 2019), estimating rates of turbulent kinetic energy dissipation in a highly stratified estuary (Lavery et al., 2013). In these cases, acoustic inversion procedures are constrained by the nature of the sampled volume, as well as the number of free parameters in the theoretical acoustic scattering model and the quality of the in-situ sampling. Water column scattering is often associated with multiple dynamics, i.e., entrained gas bubbles associated with regions of intense mixing (Marston et al., 2023). When there is no

dominant scatterer in the ensonified volume, inversion efforts are limited by the unknown combination scattering from multiple phenomena and the under-sampled nature of the inversion problem. Such issues can be overcome with knowledge of the study



site, quality ground truth data, and well-defined theoretical acoustic scattering models (e.g., Loranger et al., 2022; Lavery et al., 2010).

All considerations combined, users of broadband split-beam echosounders should be prepared to refine deployment
methodologies and employ noise reduction procedures to ensure the collection of high-quality acoustic data. Upon field work completion users should take the time to properly calibrate and correct data prior to analysis. Finally, users should carefully select and apply theoretical acoustic scattering models based on the best knowledge of data and model limitations for accurate and meaningful results.

## 6 Conclusions

Broadband split-beam echosounders are a remote sensing tool particularly well suited for field deployment in high latitude fjords. The high rate of acoustic profile collection and resulting data resolution is unmatched in comparison to traditional direct sampling. Broadband echograms provide illumination of the evolution of numerous scattering phenomena in the water column, including shear instabilities associated with bathymetry and tidally forced mixing, biological scattering from fish aggregations, and evolving thermohaline structure. As illustrated in this manuscript, these observations can be used to contextualize water
column dynamics over the full fjord scale, remotely characterize processes occurring in dangerous or inaccessible locations (e.g., the submerged terminus), and provide geophysical measurements of water column processes through broadband spectral inversion when combined with *in-situ* sampling and theoretical scattering models. Looking beyond the analysis discussed in this work, broadband split-beam systems have a history in the study of bio-physical interactions (e.g., Benoit-Bird et al., 2009; Lavery et al., 2010), seafloor emission characterization (e.g., Loranger and Weber, 2021; Weidner et al., 2019), and sea ice
(Bassett et al., 2020). The prospects for future studies of high latitude glacial fjords with broadband systems are excellent, as they can improve observational capabilities in isolated and challenging field locations through the collection of qualitative images and quantitative measurements.

Broadband split-beam echosounders are becoming increasingly available to the oceanographic community from commercial sources. This, combined with the relatively low start-up cost will increase the applicability of acoustic observations to
oceanographic research questions as processing methodologies and analysis are becoming more prevalent and are expanded to new subject areas. Acoustic observations are a relatively low-cost, low-effort addition to experimental field kits in a region where field observations are limited by length of season and generally challenging conditions. Furthermore, many research vessels, including those operating in polar regions, are already equipped with split-beam narrowband sonar systems that can be upgraded with broadband capability at minimal cost. All of this points to the increased reach of broadband echosounder
applications to polar regions, enabling an improved understanding of the complex dynamics inherent to this region through the contextual information and geophysical measurements provided by these acoustic tools.



## 7 Appendix A: Equipment field mobilization and data collection procedures

Mobilization and calibration of the broadband acoustic system and other sampling equipment on *MV Ulla Rinman* took place on July 5-6, 2023. During mobilization, the acoustic system was calibrated with a 25-mm tungsten-carbide sphere following

the well-documented procedure from Demer et al. (2015) in the western channel of Burgerbukta in Hornsund Fjord. The location was protected from swell and winds in the main channel of Hornsund fjord and was calm and ice-free during calibration operations. A CTD was taken prior to calibration to characterize the water column properties. Field calibration results were compared to a post-expedition laboratory calibration which took place in the engineering tank at the University of New Hampshire on March 13, 2024, with the same acoustic equipment. Beam pattern characterization and broadband

spectral power calibration results from the field and lab agreed, confirming the health of the transducer and calibration offset values applied in post-processing.

The Valeport miniCTD unit and the Epsi-fish microstructure probe were used for *in-situ* water column sampling and were mobilized on July 6th, 2023. The miniCTD unit was deployed from the starboard aft of *MV Ulla Rinman* on the upper level via the ship's winch. The winch provided a downward velocity of approximately 30 cm/s. The mini CTD sensor went through a

30 second initialization procedure at 1 m depth prior to the cast. The Epsi-fish was deployed from the starboard aft of *MV Ulla Rinman* via a commercial fishing reel with a controlled descent rate of ~0.6 m s$^{-1}$. Time lapse imagery was collected with two GoPro Hero11, which were deployed from the bridge deck and the transducer pole off the starboard side.

Acoustic survey activities took place between July 7 and 12, 2023 and consisted of approximately 80 km of broadband acoustic water column data, 40 CTD profiles, and 7 microstructure casts. During acoustic survey work, *MV Ulla Rinman* vessel speed

was limited to between 2-3 knots (~1-1.5 m/s), to maintain acoustic data density. Acoustic system parameters and direct sampling deployment procedures were kept consistent throughout survey activities.

The downward-looking geometry was employed to collect a series of north-south transects. The north-south transects were approximately 5 km long. All transects were run in pairs. The first transect was run continuously from the mouth of Hansbukta to the terminus of Hansbreen, collecting broadband acoustic data only. This provided continuous, uninterrupted acoustic

observations of water column dynamics at the full-fjord scale. The second transect was run in the reciprocal direction, acoustic data was collected, interspersed with ground truth data collection at specific locations: near the terminus of Hansbreen, in the glacial trough before the first set of sills, at the terminal moraine (shallowest sill), and in the main channel on Hornsund fjord at the mouth of Hansbukta. North-south transects, along with ground truth measurements, were collected in Hansbukta during multiple sections of the tidal cycle to capture the potential effect of tidal influence on mixing across the fjord. The tides in

Hornsund fjord are semi-diurnal, with two low- and high-tides of approximately equal magnitude and total tidal range generally between 2-3 m. Transect timing was determined using predicted tidal data from the Longyearbyen tidal station (https://www.tide-forecast.com/locations/Longyearbyen-Spitsbergen). In post processing, tidal phase was determined from sea surface height measured by a pressure sensor from a mooring deployed in Hansbukta during survey operations.



The side-looking geometry was utilized to scan the ice-ocean interface of Hansbreen and remotely characterize processes occurring at the terminus. The geometry of the transducer mount set-up provided a tilted beam geometry to collect observations of the near-terminus water column and the ice-ocean interface along a diagonal trajectory. The acoustic beam intersected with the submerged terminus between 15-25 m below the surface, depending on the distance between the ship and the ice face, which was not more than 200-m. The area ensonified was between 2.5 to 8.5-m². Survey lines were run from east to west, as the side-mount pole was on the starboard side of *MV Ulla Rinman*. During the survey there was minimal ice mélange, which allowed for the same parallel track to be run several times. Parallel transects included the collection of CTD profiles and time lapse photos. The CTD profile locations were chosen based on the subglacial discharge plume location, as observed in the acoustic data.

## 8 Appendix B: General flow of acoustic processing

All acoustic data presented in this manuscript are plotted as volume backscattering intensity, which has logarithm units of decibels relative to *1 mPa 1/m*. Volume backscattering intensity is a measure of the distributed scattering strength normalized by acoustic beam volume, which increases with increasing range from the transducer. Its computation is based on well-defined matched filter processing procedures (Weber and Ward, 2015; Lavery et al., 2017). The processing method has the following steps: 1) application of the match filter, 2) sampling of the acoustic time series, 3) range dependent corrections for signal attenuation from spherical spreading and absorption, 4) calibration offset and beam volume correction application, and 5) corrections for vessel motion. Steps 1-4 are described in greater detail below.

To enhance the resolution and detection of water column targets and maximize the signal-to-noise ratio (SNR), the summed signal from all four quadrants of the split-beam transducer, $s(t)$, was match filtered using an idealized transmit pulse, $s_I(t)$. The output of the match filter processing, $s_{MF}(t)$, is expressed as

$$s_{MF}(t) = \frac{s(t) \otimes s_I(t)^*}{|s_I(t)|^2}$$ (A1)

The volume backscattering intensity, $s_v$, for a specific analysis window was computed from $s_{MF}$ following a modified version of the procedure in Weber and Ward (2015)

$$s_v(t) = \frac{|s_{mf}(t)|^2}{C} \frac{r^4}{exp(-4\alpha r)} \frac{1}{V}$$ (A2)

where $r$ is the range between the region of interest and the transmitter/receiver in meters, $\alpha$ is the absorption coefficient in neper/meter evaluated for a 200 kHz signal, $C$ is the main response axis (MRA) correction factor, and $V$ is the ensonified volume. The ensonified volume can be approximated as a cylinder defined by the range from the transducer to the region of interest and the -3dB beam angle ($\theta_{3dB}$) provided by the manufacturer

$$V = \tan(\theta_{3dB}) R \frac{c\tau}{2}$$ (A3)

where $c$ is the speed of sound, $\tau$ is the pulse length in seconds, and $R$ is the range from the transducer face in meters.



In this manuscript scattering strength is reported in logarithm form of $s_v(t)$, the volume backscattering strength ($S_v$). $S_v$ is computed from $10 \log_{10} s_v$ and has units of decibels relative to *1 mPa at 1/m*.

For broadband spectra analysis, frequency-dependent scattering, $S_{mf}(f)$ was computed from the Fourier Transform of the base-banded, match-filtered output ($s_{MF}$) measured over the analysis window, $T$ (s). The ensemble average broadband spectra, $S(f)$, is then computed following

$$S(f) = \frac{\langle |S_{mf}(f)|^2 \rangle}{C(f)} \frac{r^4}{exp(-4\alpha r)} \frac{1}{V(f)} \tag{A4}$$

where the brackets $\langle \rangle$ specify an ensemble average over a number of acoustic profiles., $\alpha$ is the frequency-dependent absorption coefficient in neper/meter, $C$ is the frequency-dependent main response axis (MRA) correction factor, and $V$ is the frequency dependent ensonified area. The frequency dependent ensonified volume can be approximated as a cylinder (see Fig. 3.17 in Lurton, 2010) defined by the range from the transducer to the region of interest, and the two-way equivalent beam angle. $V$ can be calculated following (Eq. 3.44 Lurton, 2010):

$$V(f) = \Psi(f) R^2 \frac{cT}{2} \tag{A5}$$

where $T$ is the length in seconds of the analysis window, $R$ is the range to the center of the region of interest ($c(t - T/2)/2$), and $\Psi(f)$ is the frequency-dependent two-way equivalent beam angle (in steradians), as defined in Jech et al., (2005):

$$\Psi(f) = \int_{-\pi}^{\pi} \int_{0}^{\pi} b_t(\theta, \phi) b_r(\theta, \phi) \sin \theta \, d\theta \, d\phi \tag{A6}$$

where $b_t$ and $b_r$ [-] are the acoustic transmitter and receiver beam pattern respectively, and $\theta$ and $\phi$ are polar spherical coordinates.

## 9 Appendix C: Theoretical acoustic scattering model for turbulent microstructure

The wavenumber spectra for temperature and salinity are separated into three wavenumber regimes in the isotropic, homogenous turbulence model from Bachelor (1959). The measurements made in Hornsund Fjord with the ES200-7CD echosounder are limited to the viscous-convective subrange. In the viscous-convective regime, the 1D wavenumber spectra for temperature are given by Lavery et al., 2013, Eq. (2)

$$\phi_{T,S} = q \chi_{T,S} \left( \frac{\varepsilon}{\upsilon} \right)^{-1} k^{-1} \tag{A7}$$

and their co-spectra is given by:

$$\phi_{T-S} = q (\chi_T \chi_S)^{\frac{1}{2}} \left( \frac{\varepsilon}{\upsilon} \right)^{-\frac{1}{2}} k^{-1} \tag{A8}$$

where $k$ is the wavenumber ($m^{-1}$), $q$ is a dimensionless constant, defined by Oakey (1982) at $q = 3.7$, $\chi_{T,S}$ are the dissipation rates of temperature and salinity variance ($°C^2/s$, $PSU^2/s$), $\upsilon$ is the molecular viscosity ($m^2/s$), and $\varepsilon$ is the dissipation rate of turbulent kinetic energy ($m^2/s^3$).



The scattering cross section per unit volume from homogeneous and isotropic fluctuations in temperature and salinity is given by Lavery et al. 2003 (Eq. 23) as

$$\sigma_v(k) = -\frac{k^4}{K}\left(A^2\frac{\partial\phi_{T(K)}}{\partial k} + B^2\frac{\partial\phi_{S(K)}}{\partial k} + 2AB\frac{\partial\phi_{T-S(K)}}{\partial k}\right) \tag{A9}$$

A and B are terms that represent the fractional change in sound speed and density due to temperature and salinity changes. For the backscattering case in this manuscript, the wavenumber spectra in A9 are evaluated at the Bragg wavenumber ($K = 2k$).

$$\sigma_v(k) = 2^{-3}kq\left(\frac{\varepsilon}{v}\right)^{-\frac{1}{2}}\left(A^2\chi_T + B^2\chi_S + 2AB(\chi_T\chi_S)^{\frac{1}{2}}\right) \tag{A10}$$

The final expression for backscattering cross section per unit volume derived by assuming the dissipation rates of temperature and salinity are related by the mean gradients in water column properties,

$$\chi_S = \chi_T\left(\frac{\partial\bar{S}}{\partial\bar{T}}\right)^2 \tag{A11}$$

and the dissipation rates of salinity and turbulent kinetic energy can be related through the Richardson flux number ($R_F$) given by

$$R_F = \frac{g\beta}{2\varepsilon\chi_S}\left(\frac{\partial\bar{S}}{\partial z}\right)^{-1} \tag{A12}$$

where $g = 9.8\ m/s^2$ is the acceleration due to gravity and $\beta$ is the fractional change to density from salinity. Eq. (A11) and

Eq. (A12) are plugged into Eq. (A13) to provide the following expression for $\sigma_v$

$$\sigma_v(k) = 2^{-3}kq\left(\frac{\varepsilon}{v}\right)^{-\frac{1}{2}}\frac{2\varepsilon R_F}{g\beta}\frac{\partial\bar{S}}{\partial z}\left(A^2\left(\frac{\partial\bar{T}}{\partial\bar{S}}\right)^2 + B^2 + 2AB\frac{\partial\bar{T}}{\partial\bar{S}}\right) \tag{A14}$$

which is used in the analysis to match the broadband spectral measurements of the breaking internal wave.

**Data availability**

The data that support this manuscript are available at https://doi.org/10.6075/J0N87B1B as part of the University of California

San Diego Library Digital Collections.

**Author contributions**

EW and GD designed the acoustic field campaign. EW, GD, DS, HJ, HV, MC, and OG collected the dataset in Hornsund fjord. EW analysed the data with assistant from GD, AB, MF, and FS. EW prepared the manuscript with contributions from all co-authors.

**Competing interests**

The authors declare that they have no conflict of interest.



## Acknowledgements

The authors thank the captain and crew of *MV Ulla Rinman* and the staff of the Polish Polar Station in Hornsund Fjord for making the field expedition possible. Thank you to Carlo Lanzoni at the Center for Coastal and Ocean Mapping for calibrating
the acoustic equipment. We also thank Larry Mayer and Anthony Lyons at the University of New Hampshire for the loan of the broadband acoustic system used in this work. Research efforts were supported by the Office of Naval Research, Award No. N00014-23-1-2620, N00014-21-1-2304, and N00014-21-1-2316. Weidner is supported by the Scripps Institution of Oceanography postdoctoral fellowship. Vishnu and Chitre are supported by the INTERACT III Transnational Access grant under the European Union H2020 Grant Agreement No.871120. Glowacki is supported by the National Science Centre, Poland
grant no. 2021/43/D/ST10/00616.

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
