# Peer review of "High-frequency broadband active acoustic systems as a tool for highlatitude glacial fjord research"

_EGUsphere, 2024_

## Referee Comment (RC1)

"High frequency broadband acoustic systems as a tool for high latitude glacial fjord research" by Weidner et al. describes the use of echosounders to study important, and difficult to measure, hydrodynamic features in the vicinity of a tidewater glacier. While doing so it focuses on reaching new audience that could benefit from active acoustic sampling. The total amount of study-oriented information in the manuscript is limited, with most of the content focused on advocating for the broader use of broadband acoustic scattering techniques in these environments. The topics covered are interesting, relevant, and are worthy of space in the literature. However, my view is that the manuscript contains quite a bit of redundancy and that it would benefit from some restructuring.

In addition to the structural issues, there are several topics that should be revisited. Most of these are relatively minor grammatical issues, although I also believe there are several unit errors and mistakes presented in the equations that should be revisited. These, and other issues are addressed in the following itemized list of issues that need to be revisited. Other topics worth of revisiting include choices regarding the presentation of equations and the use of specific terminology when it is unnecessary. These include several processing and unit errors presented in the appendices. If these are implemented as the equations are written, then then the processing is incorrect.

My view is that the aforementioned factors collectively undermine some of the stated objectives of introducing this technology to a new audience. However, if these issues are resolved it could be effective in reaching a new audience.

**Structure recommendations:**
My fundamental concern about the structure of this paper and that it feels like there are several separate introductions and discussions. While some of these do contain different information, there is some overlapping subject matter. I think that readers would be better served by a manuscript with a long introduction, a modest "results" or "observations" section, and a long discussion section. I think that Section 4 (the descriptions of how to interpret echosounder data could go up front with a description of what echosounders and why they are useful). In Sec. 4.1 lines 239-247 could be on the intro. Similarly, I feel the initial portions of Sections 4.2 and 4.3 could, and should, be moved up. In doing so there will almost be considerable space for reducing redundancy.

**Other significant (required) revisions:**
Appendix B has several issues. I will tackle them in the order they are presented. I admit that some of these comments may be related to misinterpreting what is written so if the author's feel this is incorrect, it would be a sign that better clarifying the terms would be helpful to the readers.

First, there are several unit errors. Let's start with the units of Sv.  Following MacLennan (2005) or Medwin and Clay Sv should have units of dB re 1/m. Looking at Figure 5 I see Sv plotted two different ways and I see a different definition in terms of units on line 765.  All for the same variable. If this is simply a misunderstanding on my part, then it would be helpful to clarify these issues in the next.

Next, to the equation for Sv. I think it is acceptable to go with this approach (equation), but this choice in this context has me confused. If the purpose is to encourage more use of these methods by non-acousticians why focus on a manuscript focused on the details of seabed scattering? At a minimum the examples in the cited paper are presenting target strength and not volume backscattering.

These can be reconciled, but I wouldn't expect a non-acoustician to find this easy to navigate especially since the equations you use are not present in the cite paper.  Should you choose to continue with this I think several of would improve this section:

1. Explain how and why the equation was modified in more detail (V vs A), stating outright what volume backscattering, total backscattering cross section per unit volume, in m^2/m^3 may be helpful,
2. Similarly, defining the units and variables so that the units actual clearly balance out to 1/m,
3. Focus more on the leading denominator terms and explain its role (again, the audience will not necessarily understand, and this is a stated objective of the manuscript),
4. Note that "C," referred to here as the "main response axis-correction factor but earlier in the paper it is referred to as the calibration offset. Be consistent.

In addition, I would add sentences to other volume backscattering formulations that more clearly define how to approach in a way that is more typical to non-sea bed applications. Formulations including the specific EK80 equations often used (Andersen et al., 2024), Lavery et al. formulations, or Stanton et al. formulations would be logical choices. I think the Andersen one is one that makes the most sense because users to might be working with an EK80 would logically find those equations most common. References if fisheries acoustics (e.g., Simmonds and MacLennan also provide user friendly information). In short, I think it is okay to maintain these equations, but summarizing where this fits in and where readers can find more information to reconcile these issues makes sense in the context of this manuscript.

I also take issue with several things about the volume here.  In is well established around the community that conical representations or use of solid angle * r^2 with transducer models. This is done for the frequency domain equation, so why not reconcile these when working in the time domain as well?  They are similar but I don't think it is helpful to add to the confusion given the stated objective of the paper.

There is also, I believe, an error in the time domain representation for the broadband signal. Whether equations A3 and A5 are correct is context dependent. Those equations are specifically relevant for narrowband operations when the range resolution is driven by the pulse duration but in broadband mode the c*tau/2 used in the Appendix should be replaced with the

effectively pulse duration.  Note also that even if the narrowband approach were correct it is written differently (tau vs T) when presented.

Revisiting the unit issue: The appendix also includes at several references the units for the volume backscatter intensity that are (1) inconsistent with the documented equations and (2) inconsistent with those commonly used. Sv should have a single unit that is consistent with the literature or it should be refined. The text references units of dB re 1mPa/m. or dB re 1 mPa @ 1/m. Please make sure all of these are consistent with the definitions and other literature.

Several references in the text are NOT included in the bibliography. Please revisit the final list and make sure that these are consistent.

Line 195 and 197: The use of the term "split-beam" echosounder is used often, but it is not relevant in most places it is used. In fact, in most cases where it is used a single beam unit could easily accomplish the same thing. I recommend a word search to remove the use of "split-beam" anywhere that the discussion doesn't require a split-beam echosounder for the processing.

Line 214: The range resolution is calculated incorrect.  Assuming a sound speed of 1470 m/s and the transmitted bandwidth (80 kHz) one should calculate a theoretical range resolution of greater than 9 mm (Line 214 says 1.5 mm).  In practice this range resolution is probably less than 1 cm due to practical considerations.  Please fix.

Line 625-630:  I take issue with a few of these comments about surface noise.  Rain noise does not inherently decrease as one moves from the surface and the bonus of attenuation at shallow depths isn't going to buy much.  Going to depth with help you if your "ship noise" is indeed your ship, but that's not necessarily how I interpreted this.  What's missing here is that the electrical noise in many installations is going to be as important, if not more important, than these other issues. I strongly recommend rephrasing this.  Following onto the next page I think that much of this is unnecessary as it could be replaced with minimizing vibrations and the impacts of bubble entrained by the vessel help minimize noise and performance degradation. Notable, these transceivers/transducers have multiple stages of filtering that do manage to mitigate some of these impacts so I don't think I would dwell too much on this.

Is Section 5.2 really needed as this only includes references? I'm fine with it staying in but a few references for those unfamiliar with these issues could go in the intro and this could disappear.

**Recommended (minor) revisions:**
Below are many recommendations for other modifications to the text. Many reflect personal preferences and can be ignored while others are more substantive.

The title is missing a hyphen in "high-frequency"

I also think that you should also say echosounders or note the systems are active in the title. A high-frequency hydrophone would also be an acoustic system but isn't relevant here. You could also probably drop the "broadband" as even high-frequency narrowband systems could achieve much of what is presented here.

The list of references to acoustic scattering studies presented herein a substantial and covers a broad range of relevant literature. There are several recommendations for citations for coastal research that were not included. Examples of some relevant references that could be included are noted below in response to line 64.

Again, please cross check all references mentioned in the text and ensure they made it into the final reference list, I counted at last two references that were missing,

The word "broadband" is used too many times (in my opinion). A search for it turned up over 60 uses (excluding the references). In many of these situations the word is not helpful as the methods could explain the work. The term "broadband" could be reserved explicitly for times when there is something about the broadband operation that is required or unique. When trying to reach a broad audience it is helpful be explicit how/when/why broadband is particularly advantageous and, in my opinion, why reserving "broadband" for cases when it is relevant is important.

The following comments are take or leave, but would simplify the language and presentation.

Line 36: Strike "both in terms of sampling rate and spatial scale" and the second use of "observational"

Line 48: Strike "observational"

Line 53: The reference to Stanton and Chu here confused me some as this is typically referenced related to range resolution, but the sentence is referring to along-track length scales.

Line 56: A hydraulic jump is just one type of hydraulic transition. I recommend changing this to hydraulic transitions and note that Farmer and Amri (1989) provide a good example of the transition from sub to supercritical.

Line 64: While I would agree with the statement that echosounders are underutilized in coastal studies, there is still plenty of unreferenced examples that could have been cited here. Geyer et al (2013) [POMA] covers this matter some but misses the more modern examples that include Baschek et al 2006 (British Columbia), Kilcher and Moum 2010 (Columbia River), Geyer et al 2010 & Holleman et al 2016 (Connecticut River), and Bassett et al 2023 (James River).

Heading 1.1.

Very little of this section actual deals with "split-beam" echosounders. This is not defined and the vast majority of what is discussed is easily accomplished using single beam echosounders. I would recommend striking the split-beam in the heading but adding a sentence defining split- vs single beam systems if the goal is to reach a new audience. There is another comment about the use of split-beam references later in the manuscript.

Line 99: Recommend striking "often exceeding 500k" as I think the prior comments are adequate.

Line 106: Strike "crucial"

Line 109: Need period at end of 3)

Line 111: The sentence that starts with "Section 4" is a bit of a mouthful. Consider revising.

Line 119: The first sentence is out of place. I recommend moving to the beginning of the end paragraph where data collection is discussed.

Line 147: Caption. Strike "used in the analysis…, respectively." Refer back to this in Figs 3 and 5 instead.

Line 150: Replace "acoustic water column data" with "acoustic backscattering data throughout the water column." Then, for the next sentence, replace "Broadband acoustic water column data" with "Acoustic data"

Line 152: Replace "transmitting through a" with "with a"

Line 153: Missing a "&" in the 7CD model

Line 154: Strike "acoustic" before geometries?

156: Strike "a near-horizontal" and add "relative to the horizontal" after declination angle?

Line 157: Strike "in broadband mode" – already stated

Line 158: I don't believe the you ever mention the parameters used.

Line 160: Replace "both a" and add "s" after spheres. Also strike "well-documented"

Line 161: Is this supposed to reference Demer et al. (2015) instead?

Line 172: Note chi_t is written wrong (T is not a subscript as written)

Line 205: If the point is reach a broader audience why refer to the "along-ray path" resolution? This could be time or range (typically the vertical). Similarly, if we want to get into details, the beam ultimately diverges and will refract with the sound speed profiles so avoiding the along-ray language is probably beneficial.

Line 208: Replace "pulse bandwidth" with "transmitted signal's bandwidth"

Line 210: Replace "transducer fire" with "transducer's pulse repetition" rate (this and other recommendations are more consistent with the rest of the literature)

Figure 3. Plots need labels for units. Thev are missing in several places. In the caption an instance of SA needs a subscript. There is also a missing ")" at the end.

Line 316: Replace "draft" with "depth plus blanking distance"?

Line 319 (and many following locations): I don't think that you need to refer to the specific inset boxes in the Figures. Simply reference the figure and let the caption do the work. Looking down at the take this could clean up at least five short comments and make things cleaner.

Line 330: Rework "at the onset and ends of the sills" to just "at the ends of the sills"?

Line 335: I suppose entrained gas bubbles are possible, but that seems like an interesting hypothesis at the seabed (unless there are some seeps in which wouldn't we expect to see further bubbles rising to the surface downstream)?

Line 359: Why is this limited to high-latitude systems? Note the missing hyphen in the manuscript as well

Line 381: It is easier to just state frequency-dependent attenuation?

Line 382: I would note here that 200 m is good performance for a 200 kHz transducer installation. Many vessels struggle to get this. I'm not sure whether it is worth noting that here, but several reports on echosounder use with the ES transducers show this.

Line 438: after discharge plumes strike "such as the" and replace with "by acquiring data that reveals the"?

Line 463: "even provide remote measurements of geophysical signals." In some ways this is a bit of a stretch. Is it better to suggest that acoustic inversion may be used to infer parameters of interest (e.g., X, Y) associated with geophysical processes?

Line 475: Strike "well-known"

Line 506: Strike "broadband" as realistically this could be done with narrowband measurements?

Figure 5: Spelling error in characterize. Strike "broadband" before echogram. Unit error on the y-axis? I realize uPa are customary in acoustics, but the units here should be the same as Sv in the echogram unless a different equation is being used. If that is the case, then using a different variable name would be helpful.

Line 558: Fix unit ($m^3$)

Section 5: As stated with broadband work above, there is a lot of "high-latitude" references in here when, simply put, they aren't needed. I think it's fine to say it once, but I don't' think it needs to be repeated (used 5x in the paragraph starting at 585 alone).

Line 607: Echosounders don't need to be split beam to be calibrated. Rephrase. Echosounders also only need to be calibrated if used for quantitative purposes. There may be many in the audience here that are most interested in the qualitative (as many estuarine oceanographers are).

Line 609: I would rephrase these "corrections" as simply accounting for the underlying physics.

Line 613: Split-beam and broad again. Neither are needed. This is true of narrowband echosounders. I would rephrase this entirely as well. E.g., "Echosounders used in scientific applications have high sensitivity to scattered sound, which allows for the measurement of relatively low intensity acoustic signals."

Line 622: Strike "as well as… latitude oceans"?

Line 640: This might be the only paragraph in the paper that really requires the split-beam processing, but it doesn't say anything about them.

Line 655: This refers to Eqn A2, not 2.

Line 665: This could be simplied to "The post-processing pipeline for broadband echosounder data…" and strike "applications". The "split beam" could also be removed here

Line 670: typo, should say "processing" and split beam could be removed

Line 712: These aren't increasingly available, they are available to anyone with the finding to buy one.

Line 715: I'm not sure I agree that a WBT + split beam transducer + license is low cost (nominally $80k+ (USD) depending on what people go with?), so context may be helpful here.

Line 765-800.  See major revisions above.

Line 805: The transducer model number is correct here but is inconsistent with the prior model number provided.

Line 838: The italics in MV *Ulla Rinman* are inconsistent.

I stopped documenting split beam at some point, but I strongly recommend searching these are removing those references where they are not needed.

Lastly, I would recommend citing Bassett et al (scattering at a tidal intrusion front) somewhere in this paper. It is probably the closest analog to this work in terms of observations of scattering processes and discussions of their relevance for oceanographic studies.

---

## Referee Comment (RC2)

Manuscript Review for **High frequency broadband acoustic systems as a tool for high latitude glacial fjord research**

**Summary**

The authors describe the applications and utility of broadband active acoustics systems in glacier fjord environments to quantify important ice-ocean processes in Horsund Fjord, Svalbard. In particular, the authors observe variability of thermohaline structures and mixing, characterize a subaqueous glacier calving face, and estimate geophysical parameters. Throughout the manuscript, the authors strongly promote that broadband acoustic systems should be more widely adopted among the polar oceanography community.

**Review**

This manuscript presents a thorough description of broadband acoustic systems along with some analysis of the data collected in Horsund Fjord. Overall, this work is worthy of publication with some significant revisions. My primary concern is with the paper's structure, the amount of redundant information, and the readability. With the proper corrections, this paper could expand the audience for broadband acoustic systems for glacial fjord research.

**Significant Revisions**

There are too many introductions spread out throughout the manuscript. I realize, after reading this, that Reviewer 1 shares the same structural concern as I do. The most clear examples of this are from the first paragraph in Section 4.3 (lines 463 - 470), most of the first paragraph of Section 5 (lines 585 - 599), Section 5.1 (lines 613 - 620),  and most of Section 5.2 (640 - 670). Summarizing these sections and incorporating them into the introduction will enhance the manuscript's readability and make it more concise.

**Areas requiring clarification**

I appreciate the author's work in explaining broadband echosounder data for the non-experts. Figures 3 and 4 (particularly Figure 4) are the most significant contributions from this manuscript, in my opinion. However, I am very familiar with ice-ocean interactions, but I am not familiar with echosounder data, so Section 4 and the subsequent subsections are particularly targeted towards scientists like myself.

My main concern here is that there are clear "low-level" descriptions on how echosounders collect data, but little attention is given to how exactly to interpret Figures 3 and 4. As expanding the use of broadband acoustics is one of the main goals of this manuscript, this concern requires clarification.

What do high and low levels of Sv (dB re 1/m) mean physically? I appreciate that the authors explain the several sources of interference with the data, but providing a low-level explanation for how to interpret these units in the context of ice-ocean interactions will give readers a much better understanding of the data and the echosounders themselves.

**Line-by-line comments:**

Title: "High frequency" should be hyphenated

L10: "High frequency" should be hyphenated

L11: "high resolution" should be hyphenated

L18: "terminus" should be termini or use add an "a" before dangerous

L34: The introduction of broadband active acoustic systems comes out of nowhere here. Why are these better suited for answering outstanding questions than other methods? Introducing why broadband active acoustic systems are better suited for high-latitude coastal regions, after the second paragraph, makes this flow better and less confusing for the reader.

Put the sentence from lines 34-35 at the end of the second paragraph?

L51: "Active acoustic systems can provide these measurements." This reads very choppy. Consider removing or rephrasing.

L65: "high latitude" should be hyphenated

L106: "crucial" doesn't sound right here. Consider removing.

L109: Add a period after "estimation"

L111: "Sect. 3" to Section 3. Here and elsewhere, be consistent with what you abbreviate

L119: "Hornsund" should have "fjord" following it. This makes the language more straightforward and less colloquial. This should be corrected for other uses of "Hornsund" as well.

L121: "constitute" should be "constituting" with the current sentence structure and tense.

L125: Something is missing here. "which is among" instead of just "among"?

L126: "focused on the glaciated bay of Hansbukta" is wordy. Consider rephrasing.

L135: Having all the links in the data availability statement will make this much easier for the readers to find what data were used in this manuscript. Please remove this link and place it in the data availability statement with a description.

Figure 1: With the current layout of this figure. It makes more sense to have the top figure as "A" and the bottom figure as "B". Then, change the caption to mention the new figure labels.

The bright yellow mixing transect in the legend is hard to see. Adding a border or changing the color will help.

L155: "3D printed plate" -> "3-dimensional (3D) printed plate"

L158: Are these all the parameters?

L165: Same as the comment on line 135. Remove this link and place it in the data availability statement.

L172: Something went wrong with the latex I imagine for the "$\chi\_T$". Double check the substricpt here.

L174: Move the URL to the data availability statement here and elsewhere.

Figure 2: This is probably just me misunderstanding something, but why is there a gap between the sensor and boat in panel D? Should something be connecting the two structures?

Also, this figure caption is a tad confusing. I suggest describing each panel with its own sentence or two. For example,  (A) is the ctd used …, etc. This makes it easier for people who are simply looking at the figure to obtain the necessary information quickly.

Be consistent with your use of "3D" or "3-D"

L189: This sentence should be removed. There is no need to repeat what the section title already states.

L191: I truly appreciate the authors' efforts to engage with an audience unfamiliar with underwater acoustics. As one of my main points in "Areas requiring clarification", please expand on more of the interpretation of the data here. There is hardly any discussion on how to interpret the figures provided, namely Figures 3 and 4.

L211: "(<150-m)" Sometimes units are hyphenated and others are not. I would consider removing the hyphen here for consistency.

L221: In your list of objectives, the authors mention "geophysical parameter estimation" and now state "geophysical properties". What properties are these? Please explain here.

L223-227: "Broadband echosounder data collected in Hornsund fjord is here used to …" is awkward phrasing. I'm not sure what describing the subsections here does either. Consider removing this for consolidation and summarizing it more in the introduction.

Figure 3: Similar to Figure 2, changing the caption to have dedicated sentences for each panel will make this caption much more readable. I am also confused about the units for panels A - D. Each panel should have it's own letter, too. "(SL" needs a closing parenthesis.

L362: "In high latitude glacial fjords this remote advantage" - > In high-latitude glacial fjords, this remote advantage. There is a missing hyphen and comma.

L363: Since the authors are referring to general high-latitude glacial fjords and not a specific fjord, use an indefinite article "a" instead of "the"
"such as the submerged ice-ocean interface of the glacial terminus" -> such as a submerged ice-ocean interface of a glacial terminus

L364: "ice bergs" -> icebergs

Figure 4: Maybe I am missing something, but what exactly is the dark blue region outlined by the dashed black (?) line? Please elaborate on this in the figure caption.

L407: "strong backscattering return starting between 75 to 150 meters in range." The Y-axis only goes to a little past 120 m in range. Either correct this to 120 m or expand the Y-axis in Figure 4.

L409: "peak intensity value of between -45 to -35 dB re one μPa." This unit is different than than those shown on Figure 4 (dB re 1/m) and Figure 3. Are the authors referring to panel D in Figure 5? Please clarify.

L446-447: "With the proper deployment geometry, the broadband system  can provide direct observations of the subglacial discharge location; however, even" this sentence structure is repeated in this paragraph and used a lot in this manuscript. Consider rephrasing and varying this sentence structure for improved readability.

L448-449: "backscattering signal from the plume higher in the water column can provide important observations related to plume  dynamics." Where are the important observations the authors are referring to?

L463 - 470: As stated in the major revisions, this paragraph is not necessary here. Consider either removing it or summarizing the main points and incorporating them into the discussion.

L488: "tidal cycle (e.g.," The parentheses were never closed. Please add a closing parenthesis.

L502: "(dark gray region, Fig. 5)" Add a panel to which part of Figure 5 you are referring to

Figure 5: Please increase the thickness of the profile lines in panels A and B. It is hard to distinguish colors with such thin lines.

L543: "Taking the analysis of the acoustic measurements can be one step further". The wording of this sentence is off. Maybe "The analysis of the acoustic measurements can be expanded upon. For example…"

L564: "Challenges in applying acoustic inversion methods notwithstanding,"
This clause is awkward. Consider removing and starting the sentence with 'There is excellent...'

The authors did a nice job of explaining the limitations of acoustic inversion above; no need to reiterate here.

L577: Use future work instead of next step. This is more common and less colloquial.

L585 - 600: Reintroducing broadband echosounders is very unnecessary this late in the manuscript. Again, from my major revision, this can either be removed or summarized and included in the introduction.

L614 - 620: Same as above, this is not necessary here and better suited for the introduction.

L640 - 670: I agree with Reviewer 1. I'm not exactly sure what this section adds. Maybe summarizing the main points of this section and adding it to 5.0 would suffice.

I see the main points as 1) accurate calibration and 2) acoustic absorption.

L665 - 670: Why is this paragraph at the end of this section? Please consider moving this up in the section following the main points from above. This is the only instance where the authors explicitly mention post-processing. The order of this section in its current state is confusing. Explaining the important post-processing steps in Section 5 will succinctly summarize this section and reduce the paper length.

L675: "like those discussed in Sect. 5.2." Remove this and put (Section 5.2) and end the sentence.

L707-710: The conclusions should be about the authors' work. It is unnecessary to discuss the history of the methods from other sources here.

General comments about the conclusions: There were several references to future work in this manuscript. Placing that future work here is better for readers who skimmed the above sections. Either remove the future work references from the above sections and place them here, or briefly mention them again in this section.

---

## Referee Comment (RC3)

**Review of High frequency broadband acoustic systems as a tool for high latitude glacial fjord research**

**Summary of manuscript**

This study evaluates the use of high-frequency broadband split-beam echosounders as a tool for studying physical processes in high-latitude glacial fjords, particularly near marine-terminating glaciers. The authors demonstrate that these systems can resolve fine-scale thermohaline structure, including Kelvin-Helmholtz instabilities, detect subglacial discharge features near the ice-ocean boundary, and estimate turbulent dissipation rates using acoustic inversion techniques with good agreement to traditional instruments. The authors argue that broadband echosounders can fill critical observational gaps in fjords and serve as a low-cost, versatile tool for capturing 3D structure and mixing dynamics in challenging environments near tidewater glaciers. However, the claim that these systems can be deployed safely and at a low cost in close proximity to glacier termini without acknowledging the substantial safety constraints must be addressed, and I don't think this paper can proceed without being incorporated. It could also benefit from some reorganization for improved readability.

**Reviewer background**

My background is in collecting sonar and hydrographic data near marine-terminating glaciers. As such, my review focused primarily on the application of acoustic methods to observing processes at the glacier terminus, particularly in relation to safety, interpretation of acoustic features, and observational feasibility. I did not comment in depth on the technical implementation of the acoustic inversion algorithm or transducer signal processing methods.

**Major Revision**

The claim that high-frequency broadband echosounders are a "low-cost, low-effort addition" to experimental field kits (line 19) significantly understates the logistical and safety challenges of operating near marine-terminating glaciers. Actively calving glacier termini are extremely hazardous, and current safety guidelines typically recommend maintaining a minimum distance of at least 200–500 meters for crewed vessels depending on the glacier/location. The suggestion that this system could be routinely used in close proximity to the ice face without acknowledging these constraints is misleading and potentially dangerous. If the goal is to promote glacier-proximal observations, the authors should clearly state that such surveys must be conducted with uncrewed or remotely operated platforms to ensure safety. However, doing so would also require revising the argument about the system being low-cost and low-effort, since deploying autonomous vehicles in these environments is neither trivial nor inexpensive. This issue must be addressed directly to avoid mischaracterizing the feasibility of the method.

**Minor Revisions**

The overall clarity and readability of the manuscript could be improved by more clearly distinguishing between background, methods, results, and interpretation. For instance, the section between lines 190–210, which provides helpful background on the echosounder system, might be better suited to a dedicated background or methods section rather than appearing in "Interpretation and analysis." As a reader, I would find it clearer if the results and interpretation were more distinctly separated, or if transitions between observation and analysis were made more explicit. Additionally, some content in lines 525–540 reads more like methodological detail and could be moved to the methods or an appendix. While these changes aren't strictly necessary, they would likely strengthen the manuscript's organization and make it easier to follow.

**Specific Line Edits**

**Line 190–210:** Move the echosounder background material into a dedicated methods or background section. It currently appears in "Interpretation and analysis," which is conceptually inconsistent.

**Line 364:** "ice bergs" → "icebergs"

**Line 430:** "Generally agreed…" lacks details → Were the extents the same size? Were they located in the same area? Quantifying this agreement would strengthen the claim (e.g., "Surface expression width was within X% of the width measured acoustically").

**Section 4.3:** Consider moving this section before Section 4.2 to provide context on what geophysical parameters can be inferred from the acoustic signal before discussing plume/ice-face interpretations. I was curious as to what you think is scattering the signal within the plume after reading section 4.3. Do you think it's sediment? Can you see that from the surface expression?

**Line 489:** Missing closing parenthesis after citation.

**Line 525–540:** This section mixes methodological description with results and interpretation. Consider moving parts of this to the methods section or appendix.

**Line 565–574:** Quantify agreement between model and observation ("within a factor of 2"?), and consider plotting predicted vs. modeled.

**Figures**

**Fig 3.:**

- Consider adding scale bars to panels showing KH instabilities

**Fig 4.:**

- Can you clarify how SDP extent was determined? Was the outline based on a qualitative echogram interpretation, or was it mapped from surface ice mélange expression and transposed?
- Is it possible that ambient plume signals appear in the echogram around ~250, ~525, or ~720 m along track? Can you comment on these signatures in the main text or caption?

---

## Author Comment (AC1)

High frequency broadband acoustic systems as a tool for high latitude glacial fjord research

Response to RC2

We have reviewed the suggestions made by RC2. We thank the reviewers for the detailed review of the paper and their suggestions, in particular the reviewer's suggestions to clarify the use of acoustic data for a broader audience. We appreciate the review providing some context for sections they found confusing – in our initial submission we approached the paper as an introduction of these acoustic systems to many readers. We were glad to hear, in general our explanations provided needed context and background, but were happy to expand based on RC2's suggestions. Following their notes and suggestions, we have made changes in the manuscript using tracked changes and also provided a clean version with all edits incorporated. We have responded to each comment/suggestion below (in red), noting where we incorporated the suggestions and noting the few instances where we disagreed with the suggestion and the reasoning behind our disagreement.

**RC2 general comments**

*Summary: The authors describe the applications and utility of broadband active acoustics systems in glacier fjord environments to quantify important ice-ocean processes in Horsund Fjord, Svalbard. In particular, the authors observe variability of thermohaline structures and mixing, characterize a subaqueous glacier calving face, and estimate geophysical parameters. Throughout the manuscript, the authors strongly promote that broadband acoustic systems should be more widely adopted among the polar oceanography community.*

*Review: This manuscript presents a thorough description of broadband acoustic systems along with some analysis of the data collected in Horsund Fjord. Overall, this work is worthy of publication with some significant revisions. My primary concern is with the paper's structure, the amount of redundant information, and the readability. With the proper corrections, this paper could expand the audience for broadband acoustic systems for glacial fjord research.*

**Significant revisions**

*There are too many introductions spread out throughout the manuscript. I realize, after reading this, that Reviewer 1 shares the same structural concern as I do. The most clear examples of this are from the first paragraph in Section 4.3 (lines 463 - 470), most of the first paragraph of Section 5 (lines 585 - 599), Section 5.1 (lines 613 - 620), and most of Section 5.2 (640 - 670). Summarizing these sections and incorporating them into the introduction will enhance the manuscript's readability and make it more concise.*

We have taken the reviewers suggestion of reformatting and modified the introduction to include the preliminary discussions of high latitude systems and potential applications of acoustic measurements. These changes moved large portions of text from Sec 4 into the introduction. We modified the remaining text in Sec 4 (4.1, 4.2, 4.3) to have smooth transitions. These edits were made prior to receiving this review (RC1 had similar suggestions), but we verified the sections identified by RC2 had been accounted for in that reformatting.

**Areas requiring clarification**

I appreciate the author's work in explaining broadband echosounder data for the non-experts. Figures 3 and 4 (particularly Figure 4) are the most significant contributions from this manuscript, in my opinion. However, I am very familiar with ice-ocean interactions, but I am not familiar with echosounder data, so Section 4 and the subsequent subsections are particularly targeted towards scientists like myself.

My main concern here is that there are clear "low-level" descriptions on how echosounders collect data, but little attention is given to how exactly to interpret Figures 3 and 4. As expanding the use of broadband acoustics is one of the main goals of this manuscript, this concern requires clarification.

What do high and low levels of Sv (dB re 1/m) mean physically? I appreciate that the authors explain the several sources of interference with the data, but providing a low-level explanation for how to interpret these units in the context of ice-ocean interactions will give readers a much better understanding of the data and the echosounders themselves.

First off, we would like to thank the reviewer for highlighting their confusion in interpreting the echograms. They are absolutely correct that we are speaking to a broad audience, and we appreciate them highlighting this gap between our knowledge as a member of the underwater acoustics community and the broader oceanographic user base. We have updated several locations in the manuscript to address this concern of the reviewer in interpreting the echograms. First, we added a few sentences to Section 1.1 to note what drives acoustic scattering intensity in general – we provided some examples of typical ocean boundaries and water column phenomenon they we cn observe – references too.

We also added to Section 1.2, final paragraph, discussing relative levels of scattering intensity, noting the multiple orders of magnitude and therefore the general practice of using a logarithmic scale. Additionally, in sections 4.1 and 4.2 we added some descriptions of the scattering levels in the echograms to provide more context for readers.

**Line-by-line comments**

Title: "High frequency" should be hyphenated

Fixed.

L10: "High frequency" should be hyphenated

Fixed.

L11: "high resolution" should be hyphenated

Fixed.

L18: "terminus" should be termini or use add an "a" before dangerous

Fixed.

L34: The introduction of broadband active acoustic systems comes out of nowhere here. Why are these better suited for answering outstanding questions than other methods? Introducing why broadband active acoustic systems are better suited for high-latitude coastal regions, after the second paragraph, makes this flow better and less confusing for the reader.

This sentence is simply meant to contextualize the aim of this manuscript within the broader field and large outstanding issues facing the study of polar regions. We have modified the sentence to clarify it providing the general direction for the paper and moved it to the end of the second paragraph.

Put the sentence from lines 34-35 at the end of the second paragraph?

Implemented.

L51: "Active acoustic systems can provide these measurements." This reads very choppy. Consider removing or rephrasing.

Updated.

L65: "high latitude" should be hyphenated

Fixed.

L106: "crucial" doesn't sound right here. Consider removing.

Updated.

L109: Add a period after "estimation"

Fixed.

L111: "Sect. 3" to Section 3. Here and elsewhere, be consistent with what you abbreviate

Updated throughout the manuscript

L119: "Hornsund" should have "fjord" following it.  This makes the language more straightforward and less colloquial. This should be corrected for other uses of "Hornsund" as well.

Updated this section.

L121: "constitute" should be "constituting" with the current sentence structure and tense.

Corrected.

L125: Something is missing here. "which is among" instead of just "among"?

Corrected.

L126: "focused on the glaciated bay of Hansbukta" is wordy. Consider rephrasing.

Updated.

L135: Having all the links in the data availability statement will make this much easier for the readers to find what data were used in this manuscript. Please remove this link and place it in the data availability statement with a description.

Updated.

Figure 1: With the current layout of this figure. It makes more sense to have the top figure as "A" and the bottom figure as "B". Then, change the caption to mention the new figure labels.

The bright yellow mixing transect in the legend is hard to see. Adding a border or changing the color will help.

Updated figure.

L155: "3D printed plate" -> "3-dimensional (3D) printed plate"

Updated.

L158: Are these all the parameters?

Added frequency range to the list, this is technically a parameter that can be changed although it was not included initially since we've stated in the manuscript we are working with a certain bandwidth. Added it in the revised manuscript for clarity.

L165: Same as the comment on line 135. Remove this link and place it in the data availability statement.

Removed.

L172: Something went wrong with the latex I imagine for the "χ_T". Double check the substricpt here.

Updated.

L174: Move the URL to the data availability statement here and elsewhere.

Updated.

Figure 2: This is probably just me misunderstanding something, but why is there a gap between the sensor and boat in panel D? Should something be connecting the two structures?

This was just an issue in image rendering – updated now.

Also, this figure caption is a tad confusing. I suggest describing each panel with its own sentence or two. For example, (A) is the ctd used …, etc. This makes it easier for people who are simply looking at the figure to obtain the necessary information quickly.

Updated.

Be consistent with your use of "3D" or "3-D"

Checked throughout.

L189: This sentence should be removed. There is no need to repeat what the section title already states.

Removed.

L191: I truly appreciate the authors' efforts to engage with an audience unfamiliar with underwater acoustics. As one of my main points in "Areas requiring clarification", please expand on more of the interpretation of the data here. There is hardly any discussion on how to interpret the figures provided, namely Figures 3 and 4.

See our response to major comments above – we have expanded this section and the section on Figure 4 to help readers interpret echogram figures. Again, we thank the reviewer for pointing this out – very helpful comment from an audience member not deeply familiar with echosounder acoustics!

L211: "(<150-m)" Sometimes units are hyphenated and others are not. I would consider removing the hyphen here for consistency.

Updated all unit measurements throughout manuscript for consistency.

L221: In your list of objectives, the authors mention "geophysical parameter estimation" and now state "geophysical properties". What properties are these? Please explain here.

Updated this with some examples, also this section has been reworked so no appears in the introduction.

L223-227: "Broadband echosounder data collected in Hornsund fjord is here used to ..." is awkward phrasing. I'm not sure what describing the subsections here does either. Consider removing this for consolidation and summarizing it more in the introduction.

This suggestion has been implemented and moved to the introduction.

Figure 3: Similar to Figure 2, changing the caption to have dedicated sentences for each panel will make this caption much more readable. I am also confused about the units for panels A - D. Each panel should have it's own letter, too. "(SL" needs a closing parenthesis.

Updated with suggestions.

L362: "In high latitude glacial fjords this remote advantage" - > In high-latitude glacial fjords, this remote advantage. There is a missing hyphen and comma.

Updated, this section was moved up to the introduction.

L363: Since the authors are referring to general high-latitude glacial fjords and not a specific fjord, use an indefinite article "a" instead of "the" "such as the submerged ice-ocean interface of the glacial terminus" -> such as a submerged ice-ocean interface of a glacial terminus

Updated.

L364: "ice bergs" -> icebergs

Updated.

Figure 4: Maybe I am missing something, but what exactly is the dark blue region outlined by the dashed black (?) line? Please elaborate on this in the figure caption.

The figure caption noted the black dashed line highlights the terminus. We added arrows to the figure to point from the label in the figure "ice-ocean interface" to the scattering region.

L407: "strong backscattering return starting between 75 to 150 meters in range." The Y-axis only goes to a little past 120 m in range. Either correct this to 120 m or expand the Y-axis in Figure 4.

Updated.

L409: "peak intensity value of between -45 to -35 dB re one µPa." This unit is different than than those shown on Figure 4 (dB re 1/m) and Figure 3. Are the authors referring to panel D in Figure 5? Please clarify.

Updated the manuscript to remove these values.

L446-447: "With the proper deployment geometry, the broadband system  can provide direct observations of the subglacial discharge location; however, even" this sentence structure is repeated in this paragraph and used a lot in this manuscript. Consider rephrasing and varying this sentence structure for improved readability.

Updated.

L448-449: "backscattering signal from the plume higher in the water column can provide important observations related to plume dynamics." Where are the important observations the authors are referring to?

Updated.

L463 - 470: As stated in the major revisions, this paragraph is not necessary here. Consider either removing it or summarizing the main points and incorporating them into the discussion.

This has been updated.

L488: "tidal cycle (e.g.," The parentheses were never closed. Please add a closing parenthesis.

Updated.

L502: "(dark gray region, Fig. 5)" Add a panel to which part of Figure 5 you are referring to

Removed the reference to the "dark gray region"

Figure 5: Please increase the thickness of the profile lines in panels A and B. It is hard to distinguish colors with such thin lines.

Updated.

L543: "Taking the analysis of the acoustic measurements can be one step further". The wording of this sentence is off. Maybe "The analysis of the acoustic measurements can be expanded upon. For example…"

Great suggestion, implemented.

L564: "Challenges in applying acoustic inversion methods notwithstanding," This clause is awkward. Consider removing and starting the sentence with 'There is excellent…' The authors did a nice job of explaining the limitations of acoustic inversion above; no need to reiterate here.

Implemented

L577: Use future work instead of next step. This is more common and less colloquial.

Updated.

L585 - 600: Reintroducing broadband echosounders is very unnecessary this late in the manuscript. Again, from my major revision, this can either be removed or summarized and included in the introduction.

Removed this paragraph entirely.

L614 - 620: Same as above, this is not necessary here and better suited for the introduction.

Here we believe an in depth discussion of receiver sensitivity would slow the pace of the introduction; hence it belongs in its own section for those readers who are considering using these systems in the future. Outlining the importance of quality data collection requires its own section.

However, we do understand this section, as originally written is overly long and we have tightened it up.

L640 - 670: I agree with Reviewer 1. I'm not exactly sure what this section adds. Maybe summarizing the main points of this section and adding it to 5.0 would suffice.

I see the main points as 1) accurate calibration and 2) acoustic absorption.

Tightened up this section.

L665 - 670: Why is this paragraph at the end of this section? Please consider moving this up in the section following the main points from above. This is the only instance where the authors explicitly mention post-processing. The order of this section in its current state is confusing. Explaining the important post-processing steps in Section 5 will succinctly summarize this section and reduce the paper length.

Moved this section up.

L675: "like those discussed in Sect. 5.2." Remove this and put (Section 5.2) and end the sentence.

Updated.

L707-710: The conclusions should be about the authors' work. It is unnecessary to discuss the history of the methods from other sources here.

We feel highlighting other potential analyses are important for a new audience, especially in the conclusion to reiterate that there are even more applications out there for these methods, given the aim of this manuscript is to introduce this tool to a broad audience, and not just highlight the work we have done using this system.

General comments about the conclusions: There were several references to future work in this manuscript. Placing that future work here is better for readers who skimmed the above sections. Either remove the future work references from the above sections and place them here, or briefly mention them again in this section.

Added.

---

## Author Comment (AC2)

High frequency broadband acoustic systems as a tool for high latitude glacial fjord research

Response to RC1

We have reviewed the suggestions made by RC1. We thank the reviewer for the detailed review of the paper and their suggestions, particularly in clarifying the acoustic processing for a broader audience and updating the paper structure. We feel the paper is significantly tightened, with a better flow. Following RC1's notes and suggestions, we have made changes in the manuscript using tracked changes and will also provide a clean version with all edits incorporated. We have responded to each comment/suggestion below (in red), noting where we incorporated the suggestions and noting the few instances where we disagreed with the suggestion and the reasoning behind our disagreement.

**RC1 general comments:**

*"High frequency broadband acoustic systems as a tool for high latitude glacial fjord research" by Weidner et al. describes the use of echosounders to study important, and difficult to measure, hydrodynamic features in the vicinity of a tidewater glacier. While doing so it focuses on reaching new audience that could benefit from active acoustic sampling. The total amount of study-oriented information in the manuscript is limited, with most of the content focused on advocating for the broader use of broadband acoustic scattering techniques in these environments. The topics covered are interesting, relevant, and are worthy of space in the literature. However, my view is that the manuscript contains quite a bit of redundancy and that it would benefit from some restructuring.*

*In addition to the structural issues, there are several topics that should be revisited. Most of these are relatively minor grammatical issues, although I also believe there are several unit errors and mistakes presented in the equations that should be revisited. These, and other issues are addressed in the following itemized list of issues that need to be revisited. Other topics worth of revisiting include choices regarding the presentation of equations and the use of specific terminology when it is unnecessary. These include several processing and unit errors presented in the appendices. If these are implemented as the equations are written, then then the processing is incorrect.*

*My view is that the aforementioned factors collectively undermine some of the stated objectives of introducing this technology to a new audience. However, if these issues are resolved it could be more effective in reaching a new audience.*

*The attached supplemental information has details regarding the recommended scope of the revision.*

**Structure recommendations**:

*My fundamental concern about the structure of this paper and that it feels like there are several separate introductions and discussions. While some of these do contain different information, there is some overlapping subject matter. I think that readers would be better served by a manuscript with a long introduction, a modest "results" or "observations" sec on, and a long discussion section. I think that Sec on 4 (the descriptions of how to interpret echosounder data could go up front with a description of what echosounders and why they are useful). In Sec. 4.1 lines 239-247 could be on the intro. Similarly, I feel the initial portions of Sections 4.2 and 4.3 could, and should, be moved up. In doing so there will almost be considerable space for reducing redundancy.*

We have taken RC1's suggestion of reformatting and modified the introduction to include the preliminary discussions of high latitude systems and potential applications of acoustic measurements. These changes moved large portions of text from Sec 4 into the introduction. We modified the remaining text in Sec 4 (4.1, 4.2, 4.3) to have smooth transitions. We also tightened these sections up to help with flow and paper clarity.

**Other significant (required) revisions**:

*Appendix B has several issues. I will tackle them in the order they are presented. I admit that some of these comments may be related to misinterpreting what is written so if the author's feel this is incorrect, it would be a sign that be er clarifying the terms would be helpful to the readers.*

*First, there are several unit errors. Let's start with the units of Sv. Following MacLennan (2005) or Medwin and Clay Sv should have units of dB re 1/m. Looking at Figure 5 I see Sv plotted two different ways and I see a different definition in terms of units on line 765. All for the same variable. If this is simply a misunderstanding on my part, then it would be helpful to clarify these issues in the next.*

We have updated both the Sv units in Figure 5 and on line 765 – as noted by the reviewer these should all have the same units, and typos from preliminary processing and figure design persisted into the submitted version.

*Next, to the equation for Sv. I think it is acceptable to go with this approach (equation), but this choice in this context has me confused. If the purpose is to encourage more use of these methods by non-acousticians why focus on a manuscript focused on the details of seabed scattering? At a minimum the examples in the cited paper are presenting target strength and not volume backscattering.*

The reviewer makes an excellent point regarding the Weber and Ward reference and its relation to Eq. 2 – the reason be focused in on this particular reference was the thorough derivation of the cited equation, which we have found useful in developing a general understanding of acoustic corrections for attenuation and beam pattern effects. However, the reviewer's point that this is a paper focused on seabed scattering and therefore an area correction, rather than a volume correction, is fair. We did not provide a clear process for modification of the referenced equation.

*These can be reconciled, but I wouldn't expect a non-acoustician to find this easy to navigate especially since the equations you use are not present in the cite paper. Should you choose to continue with this I think several of would improve this sec on:*

*1. Explain how and why the equation was modified in more detail (V vs A), stating outright what volume backscattering, total backscattering cross sec on per unit volume, in m^2/m^3 may be helpful,*

See below.

*2. Similarly, defining the units and variables so that the units actual clearly balance out to 1/m,*

See below.

*3. Focus more on the leading denominator terms and explain its role (again, the audience will not necessarily understand, and this is a stated objective of the manuscript),*

See below.

*4. Note that "C," referred to here as the "main response axis-correct on factor but earlier in the paper it is referred to as the calibration offset. Be consistent.*

See below.

*In addition, I would add sentences to other volume backscattering formulations that more clearly define how to approach in a way that is more typical to non-seabed applications. Formulations including the specific EK80 equations often used (Andersen et al., 2024), Lavery et al. formulations, or Stanton et al. formulations would be logical choices. I think the Andersen one is one that makes the most sense because users to might be working with an EK80 would logically find those equations most common. References if fisheries acoustics (e.g., Simmonds and MacLennan also provide user friendly information). In short, I think it is okay to maintain these equations, but summarizing where this fits in and where readers can find more information to reconcile these issues makes sense in the context of this manuscript.*

We have modified Appendix B following the suggestions 1-3 below to clarify how we modified the Weber and Ward seafloor scattering equation to a volume scattering equation. We have corrected the definition for "C" throughout the manuscript and we also point readers toward Andersen et al., 2024 and Lavery et al. (2017) for alternative formations of volume scattering – particularly suited to biological applications.

*I also take issue with several things about the volume here. In is well established around the community that conical representations or use of solid angle * r^2 with transducer models. This is done for the frequency domain equation, so why not reconcile these when working in the time domain as well? They are similar but I don't think it is helpful to add to the confusion given the stated objective of the paper.*

The reviewer is correct that we could use the solid angle volume calculation for the time domain Sv volume corrections. We opted to use the simpler cylinder estimation calculation for the time domain work based on the published -3 dB beam angle since the time domain volume calculation doesn't need to account for frequency-driven changes in beam angle. For new users this is likely more accessible than the computation of the f-dependent two-way equivalent beam angle. We added a note at the end of the section that a two-way equivalent beam angle could also be used to compute volume for a time-domain correction, so that new users understand this could be done (but is not required).

*There is also, I believe, an error in the time domain representation for the broadband signal. Whether equations A3 and A5 are correct is context dependent. Those equations are specifically relevant for narrowband operations when the range resolution is driven by the pulse duration but in broadband mode the c*tau/2 used in the Appendix should be replaced with the effectively pulse duration. Note also that even if the narrowband approach were correct it is written differently (tau vs T) when presented.*

We have clarified A3 – the time domain volume calculation, tau here was the effective pulse duration but failed to note this and this has been spelled out.

We are uncertain if the reviewer had an issue with A5. A5 is the volume correction in the frequency domain for a selected sub-section of an acoustic profile (analysis window) and therefore, should not use the effective pulse duration but the length of the analysis window. This would provide a volume correction for analysis windows of different size, whereas using the effective pulse duration would correct for different size analysis windows leading to an issue in comparison at different ranges.

*Revisiting the unit issue: The appendix also includes at several references the units for the volume backscatter intensity that are (1) inconsistent with the documented equations and (2) inconsistent with those commonly used. Sv should have a single unit that is consistent with the literature or it should be refined. The text references units of dB re 1mPa/m. or dB re 1 mPa @ 1/m. Please make sure all of these are consistent with the definitions and other literature.*

As noted above, Sv units have been reassessed throughout the manuscript and are in line with the referenced material.

*Several references in the text are NOT included in the bibliography. Please revisit the final list and make sure that these are consistent. Line 195 and 197: The use of the term "split-beam" echosounder is used often, but it is not relevant in most places it is used. In fact, in most cases where it is used a single beam unit could easily accomplish the same thing. I recommend a word search to remove the use of "splitbeam" anywhere that the discussion doesn't require a split-beam echosounder for the processing.*

References have been reviewed and those missing added to the reference list.

Additionally, the use of split-beam throughout the manuscript was reviewed and removed in locations where split-aperture processing was unnecessary. Just noting here, the term split-beam was used to be consistent across the manuscript in referring to the system in an effort to reach a broader audience (reduce confusion). We acknowledge that in most cases we are not actually using split aperture processing for much of this analysis and have instead incorporated the use of scientific echosounder for much of the manuscript.

*Line 214: The range resolution is calculated incorrect. Assuming a sound speed of 1470 m/s and the transmitted bandwidth (80 kHz) one should calculate a theoretical range resolution of greater than 9 mm (Line 214 says 1.5 mm). In practice this range resolution is probably less than 1 cm due to practical considerations. Please fix.*

Thank you for catching this, range resolution has been recalculated and edited.

*Line 625-630: I take issue with a few of these comments about surface noise. Rain noise does not inherently decrease as one moves from the surface and the bonus of attenuation at shallow depths isn't going to buy much. Going into depth with help you if your "ship noise" is indeed your ship, but that's not necessarily how I interpreted this. What's missing here is that the electrical noise in many installations is going to be as important, if not more important, than these other issues. I strongly recommend rephrasing this. Following onto the next page I think that much of this is unnecessary as it could be replaced with minimizing vibrations and the impacts of bubbles entrained by the vessel help minimize noise and performance degradation. Notable, these transceivers/transducers have multiple stages of filtering that do manage to mitigate some of these impacts so I don't think I would dwell too much on this.*

We have updated this section to reflect these suggestions – in particular identifying electrical interference as the primary source of noise.

*Is Sec on 5.2 really needed as this only includes references? I'm fine with it staying in but a few references for those unfamiliar with these issues could go in the intro and this could disappear.*

We included the step-by-step derivation for Eq. 1 to support understanding by new users and those familiar with turbulence literature. Certainly, these readers could refer to Lavery or Muchowski but we thought providing the derivation here was more straightforward.

**Recommended (minor) revisions:**

*Below are many recommendations for other modifications to the text. Many reflect personal preferences and can be ignored while others are more substantive.*

*The title is missing a hyphen in "high-frequency"*

Updated.

*I also think that you should also say echosounders or note the systems are active in the title. A high-frequency hydrophone would also be an acoustic system but isn't relevant here. You could also probably drop the "broadband" as even high-frequency narrowband systems could achieve much of what is presented here.*

Clarified that we were referring to active acoustic systems, but we reatin broadband in the title as the broadband nature of the data allows for the high range resolution and the spectral analysis.

*The list of references to acoustic scattering studies presented herein a substantial and covers a broad range of relevant literature. There are several recommendations for citations for coastal research that were not included. Examples of some relevant references that could be included are noted below in response to line 64.*

Added these references in. Thank you for pointing them out.

*Again, please cross check all references mentioned in the text and ensure they made it into the final reference list, I counted at last two references that were missing,*

References have been reviewed and the missing references have been added to the manuscript.

*The word "broadband" is used too many times (in my opinion). A search for it turned up over 60 uses (excluding the references). In many of these situations the word is not helpful as the methods could explain the work. The term "broadband" could be reserved explicitly for times when there is something about the broadband operation that is required or unique. When trying to reach a broad audience it is helpful be explicit how/when/why broadband is particularly advantageous and, in my opinion, why reserving "broadband" for cases when it is relevant is important.*

Reviewed the manuscript and removed "broadband" sentences where it was unnecessary.

**The following comments are take or leave, but would simplify the language and presentation:**

*Line 36: Strike "both in terms of sampling rate and spatial scale" and the second use of "observational"*

Noted and edited.

*Line 48: Strike "observational"*

The sentence seems too vague without noting the "gap" we are discussing is observations.

*Line 53: The reference to Stanton and Chu here confused me some as this is typically referenced related to range resolution, but the sentence is referring to along-track length scales.*

Noted and edited.

*Line 56: A hydraulic jump is just one type of hydraulic transition. I recommend changing this to hydraulic transitions and note that Farmer and Amri (1989) provide a good example of the transition from sub to supercritical.*

Changed jump to transition, but did not include the specifics about the Farmer and Amri work, since we are just noting the general process that can be observed.

*Line 64: While I would agree with the statement that echosounders are underutilized in coastal studies, there is still plenty of unreferenced examples that could have been cited here. Geyer et al (2013) [POMA] covers this ma er some but misses the more modern examples that include Baschek et al 2006 (British Columbia), Kilcher and Moum 2010 (Columbia River), Geyer et al 2010 & Holleman et al 2016 (Connecticut River), and Bassett et al 2023 (James River).*

Added Kilcher and Nash (2010), Lavery et al. (2013), and Bassett et al. (2023). Holleman does not center their analysis with acoustic data, rather references Lavery et al., 2013.

*Heading 1.1. Very little of this section actual deals with "split-beam" echosounders. This is not defined and the vast majority of what is discussed is easily accomplished using single beam echosounders. I would recommend striking the split-beam in the heading but adding a sentence defining split- vs single beam systems if the goal is to reach a new audience. There is another comment about the use of split-beam references later in the manuscript.*

We added a definition of split-beam echosounder, as a type of scientific echosounder, into the section. But we did not remove split-beam from the heading title, as the acoustic system used here was a split-beam, not a single-beam. We did rely on split-aperture processing for the calibration, so the identification matters. We did remove (as requested in later comments) the references to split-beam from locations where single beam systems could be used.

*Line 99: Recommend striking "often exceeding 500k" as I think the prior comments are adequate.*

We added the cost to clarify for the reader what "high cost" means in this context. Different groups could have very different views of what constitutes an expensive system.

*Line 106: Strike "crucial"*

Noted and edited.

*Line 109: Need period at end of 3)*

Noted and edited.

*Line 111: The sentence that starts with "Sec on 4" is a bit of a mouthful. Consider revising.*

Noted and edited.

*Line 119: The first sentence is out of place. I recommend moving to the beginning of the end paragraph where data collection is discussed.*

Noted and edited.

*Line 147: Cap on. Strike "used in the analysis..., respectively." Refer back to this in Figs 3 and 5 instead.*

Noted and edited.

*Line 150: Replace "acoustic water column data" with "acoustic backscattering data throughout the water column." Then, for the next sentence, replace "Broadband acoustic water column data" with "Acoustic data"*

Noted and edited.

*Line 152: Replace "transmitting through a" with "with a"*

Noted and edited.

*Line 153: Missing a "&" in the 7CD model*

Guessing you mean "7", not "&", noted and edited.

*Line 154: Strike "acoustic" before geometries?*

Noted and edited.

*Line 156: Strike "a near-horizontal" and add "relative to the horizontal" after declination angle?*

Noted and edited.

*Line 157: Strike "in broadband mode" – already stated*

Noted and edited.

*Line 158: I don't believe the you ever mention the parameters used.*

*Added to the appendix, which is noted in the manuscript on line 177.*

*Line 160: Replace "both a" and add "s" after spheres. Also strike "well-documented"*

Noted and edited.

*Line 161: Is this supposed to reference Demer et al. (2015) instead?*

Yes.

*Line 172: Note chi_t is written wrong (T is not a subscript as written)*

Noted and edited.

*Line 205: If the point is reach a broader audience why refer to the "along-ray path" resolution? This could be me or range (typically the vertical). Similarly, if we want to get into details, the beam*

*ultimately diverges and will refract with the sound speed profiles so avoiding the alongray language is probably beneficial.*

Point taken, although vertical range resolution is not appropriate for this paper given we use a tilted geometry in the submerged ice face/subglacial discharge plume section and vertical range resolution wouldn't be appropriate.

*Line 208: Replace "pulse bandwidth" with "transmitted signal's bandwidth"*

Noted and edited.

*Line 210: Replace "transducer fire" with "transducer's pulse repetition" rate (this and other recommendations are more consistent with the rest of the literature)*

Noted and edited.

*Figure 3. Plots need labels for units. They are missing in several places. In the caption an instance of SA needs a subscript. There is also a missing ")" at the end.*

Noted and edited.

*Line 316: Replace "draft" with "depth plus blanking distance"?*

Noted and edited.

*Line 319 (and many following locations): I don't think that you need to refer to the specific inset boxes in the Figures. Simply reference the figure and let the cap on do the work. Looking down at the take this could clean up at least five short comments and make things cleaner.*

Noted and edited.

*Line 330: Rework "at the onset and ends of the sills" to just "at the ends of the sills"?*

Reworked to "at either end of the sills".

*Line 335: I suppose entrained gas bubbles are possible, but that seems like an interesting hypothesis at the seabed (unless there are some seeps in which wouldn't we expect to see further bubbles rising to the surface downstream)?*

This is a fair point for the deep K-H billows near the seabed – there is not a clear formation mechanism. We'll strike that suggestion of gas, given the origin of the gas (or the fate of the bubbles) hasn't been observed/can't easily be explained.

*Line 359: Why is this limited to high-latitude systems? Note the missing hyphen in the manuscript as well*

Noted and edited.

*Line 381: It is easier to just state frequency-dependent attenuation?*

Certainly, we agree, and this suggestion would be appropriate in a paper for an acoustics audience who immediately would understand what frequency-dependent attenuation is and what drives it. Here we thought more clarity on the process would be better.

*Line 382: I would note here that 200 m is good performance for a 200 kHz transducer installation. Many vessels struggle to get this. I'm not sure whether it is worth noting that here, but several reports on echosounder use with the ES transducers show this.*

We removed the discussion of maximum range of the ES200 system in the reformatting of the paper structure.

*Line 438: after discharge plumes strike "such as the" and replace with "by acquiring data that reveals the"?*

Noted and edited.

*Line 463: "even provide remote measurements of geophysical signals." In some ways this is a bit of a stretch. Is it better to suggest that acoustic inversion may be used to infer parameters of interest (e.g., X, Y) associated with geophysical processes?*

This seems like an issue of semantics, parameters of interest associated with geophysical processes could be described as geophysical signals. We did add in the examples - (e.g. X, Y) - to clarify.

*Line 475: Strike "well-known"*

Noted and edited.

*Line 506: Strike "broadband" as realistically this could be done with narrowband measurements?*

Since this is a description of how we went about the measurements, using broadband data rather than narrowband, we are leaving broadband in the sentence on line 506. We do agree that a narrowband system could be used to make similar inversion measurements although the spectral analysis steps would likely not be useful and the range resolution would be reduced.

*Figure 5: Spelling error in characterize. Strike "broadband" before echogram.*

Noted and edited.

*Figure 5: Unit error on the y-axis? I realize uPa are customary in acoustics, but the units here should be the same as Sv in the echogram unless a different equation is being used. If that is the case, then using a different variable name would be helpful.*

Noted and edited, see major revisions section above.

*Line 558: Fix unit (m3)*

Noted and edited.

*Section 5: As stated with broadband work above, there is a lot of "high-latitude" references in here when, simply put, they aren't needed. I think it's fine to say it once, but I don't' think it needs to be repeated (used 5x in the paragraph starting at 585 alone).*

Removed high-latitude from several sentences where it was erroneous

*Line 607: Echosounders don't need to be split beam to be calibrated. Rephrase. Echosounders also only need to be calibrated if used for quantitative purposes. There may be many in the audience here that are most interested in the qualitative (as many estuarine oceanographers are).*

We agree with the reviewer that 1) you can calibrate a single beam echosounder and 2) that if the application is not quantitative, calibration is not needed. However, the sentence in question states, "Second, quantitative analysis, such as broadband spectral characterization or acoustic inversion procedures, require calibration...", so we are specifically talking about circumstances, as you note, where calibration is required. Furthermore, although calibration does not always require split-beam systems, the calibration of the system we are using (and the references we refer to) does use split aperture processing to position the sphere to correct for beam pattern effects.

*Line 609: I would rephrase these "corrections" as simply accounting for the underlying physics.*

Noted and edited.

*Line 613: Split-beam and broad again. Neither are needed. This is true of narrowband echosounders. I would rephrase this entirely as well. E.g., "Echosounders used in scientific applications have high sensitivity to scattered sound, which allows for the measurement of relatively low intensity acoustic signals."*

Noted that both the narrowband and broadband applications of scientific echosounders have the same sensitivity. To be clear, our point was that SNR of broadband systems is higher than then narrowband counterparts and broadband systems are more sensitive to external noises because their receive sensitivity covers a broader bandwidth than a narrowband system. We are modified the sentences to make this clearer and removed the unnecessary split-beam/broadband words where applicable.

*Line 622: Strike "as well as... latitude oceans"?*

Noted and edited.

*Line 640: This might be the only paragraph in the paper that really requires the split-beam processing, but it doesn't say anything about them.*

Added a specific note about the use of split-aperture processing to determine the position of the sphere. We also note the calibration method is that of a split-beam system (and we reference Demer).

*Line 655: This refers to Eqn A2, not 2.*

Noted and edited.

*Line 665: This could be simplified to "The post-processing pipeline for broadband echosounder data..." and strike "applications". The "split beam" could also be removed here*

Noted and edited.

*Line 670: typo, should say "processing" and split beam could be removed*

Noted and edited.

*Line 712: These aren't increasingly available, they are available to anyone with the finding to buy one.*

Noted and edited.

*Line 715: I'm not sure I agree that a WBT + split beam transducer + license is low cost (nominally $80k+ (USD) depending on what people go with?), so context may be helpful here.*

Expanded sentence to include specific cost (USD/EUR) of the system. We do note the start up is *relatively low cost* and hope by specifying the amount the reader can make their own determination on whether this is low. We would argue that given the cost of Arctic research this is a low-cost system. For example, the daily cost of the leased ship was effectively equivalent to the price of the combined system.

*Line 765-800.  See major revisions above.*

*See response to major revisions above.*

*Line 805: The transducer model number is correct here but is inconsistent with the prior model number provided.*

Corrected model number in section 3 and verified model number is correct elsewhere.

*Line 838: The italics in MV Ulla Rinman are inconsistent.*

Fixed.

*I stopped documenting split beam at some point, but I strongly recommend searching these are removing those references where they are not needed.*

We reviewed the manuscript and removed a number of references to split beam, sometimes removing entirely or replacing with scientific.

*Lastly, I would recommend citing Bassett et al (scattering at a tidal intrusion front) somewhere in this paper. It is probably the closest analog to this work in terms of observations of scattering processes and discussions of their relevance for oceanographic studies.*

Cited on line 64.

**RC2 general comments**

*Summary: The authors describe the applications and utility of broadband active acoustics systems in glacier fjord environments to quantify important ice-ocean processes in Horsund Fjord, Svalbard. In particular, the authors observe variability of thermohaline structures and mixing, characterize a subaqueous glacier calving face, and estimate geophysical parameters. Throughout the manuscript, the authors strongly promote that broadband acoustic systems should be more widely adopted among the polar oceanography community.*

*Review: This manuscript presents a thorough description of broadband acoustic systems along with some analysis of the data collected in Horsund Fjord. Overall, this work is worthy of publication with some significant revisions. My primary concern is with the paper's structure, the amount of redundant information, and the readability. With the proper corrections, this paper could expand the audience for broadband acoustic systems for glacial fjord research.*

**Significant revisions**

*There are too many introductions spread out throughout the manuscript. I realize, after reading this, that Reviewer 1 shares the same structural concern as I do. The most clear examples of this are from the first paragraph in Section 4.3 (lines 463 - 470), most of the first paragraph of Section 5 (lines 585 - 599), Section 5.1 (lines 613 - 620), and most of Section 5.2 (640 - 670). Summarizing these sections and incorporating them into the introduction will enhance the manuscript's readability and make it more concise.*

We have taken the reviewers suggestion of reformatting and modified the introduction to include the preliminary discussions of high latitude systems and potential applications of acoustic measurements. These changes moved large portions of text from Sec 4 into the introduction. We modified the remaining text in Sec 4 (4.1, 4.2, 4.3) to have smooth transitions.

**Areas requiring clarification**

I appreciate the author's work in explaining broadband echosounder data for the non-experts. Figures 3 and 4 (particularly Figure 4) are the most significant contributions from this manuscript, in my opinion. However, I am very familiar with ice-ocean interactions, but I am not familiar with echosounder data, so Section 4 and the subsequent subsections are particularly targeted towards scientists like myself.

My main concern here is that there are clear "low-level" descriptions on how echosounders collect data, but little attention is given to how exactly to interpret Figures 3 and 4. As expanding the use of broadband acoustics is one of the main goals of this manuscript, this concern requires clarification.

What do high and low levels of Sv (dB re 1/m) mean physically? I appreciate that the authors explain the several sources of interference with the data, but providing a low-level explanation for how to interpret these units in the context of ice-ocean interactions will give readers a much better understanding of the data and the echosounders themselves.

First off, we would like to thank the reviewer for highlighting their confusion in interpreting the echograms. They are absolutely correct that we are speaking to a broad audience, and we

appreciate them highlighting this gap between our knowledge as a member of the underwater acoustics community and the broader oceanographic user base. We have updated several locations in the manuscript to address this concern of the reviewer in interpreting the echograms. First, we added a few sentences to Section 1.1 to note what drives acoustic scattering intensity in general – we provided some examples of typical ocean boundaries and water column phenomenon they we cn observe – references too.

We also added to Section 1.2, final paragraph, discussing relative levels of scattering intensity, noting the multiple orders of magnitude and therefore the general practice of using a logarithmic scale.

Additionally, in sections 4.1 and 4.2 we added some descriptions of the scattering levels in the echograms to provide more context for readers.

**Line-by-line comments**

Title: "High frequency" should be hyphenated

Fixed.

L10: "High frequency" should be hyphenated

Fixed.

L11: "high resolution" should be hyphenated

Fixed.

L18: "terminus" should be termini or use add an "a" before dangerous

Fixed.

L34: The introduction of broadband active acoustic systems comes out of nowhere here. Why are these better suited for answering outstanding questions than other methods? Introducing why broadband active acoustic systems are better suited for high-latitude coastal regions, after the second paragraph, makes this flow better and less confusing for the reader.

This sentence is simply meant to contextualize the aim of this manuscript within the broader field and large outstanding issues facing the study of polar regions. We have modified the sentence to clarify it providing the general direction for the paper and moved it to the end of the second paragraph.

Put the sentence from lines 34-35 at the end of the second paragraph?

Implemented.

L51: "Active acoustic systems can provide these measurements." This reads very choppy. Consider removing or rephrasing.

Updated.

L65: "high latitude" should be hyphenated

Fixed.

L106: "crucial" doesn't sound right here. Consider removing.

Updated.

L109: Add a period after "estimation"

Fixed.

L111: "Sect. 3" to Section 3. Here and elsewhere, be consistent with what you abbreviate

Updated throughout the manuscript

L119: "Hornsund" should have "fjord" following it.  This makes the language more straightforward and less colloquial. This should be corrected for other uses of "Hornsund" as well.

Updated this section.

L121: "constitute" should be "constituting" with the current sentence structure and tense.

Corrected.

L125: Something is missing here. "which is among" instead of just "among"?

Corrected.

L126: "focused on the glaciated bay of Hansbukta" is wordy. Consider rephrasing.

Updated.

L135: Having all the links in the data availability statement will make this much easier for the readers to find what data were used in this manuscript. Please remove this link and place it in the data availability statement with a description.

Updated.

Figure 1: With the current layout of this figure. It makes more sense to have the top figure as "A" and the bottom figure as "B". Then, change the caption to mention the new figure labels.

The bright yellow mixing transect in the legend is hard to see. Adding a border or changing the color will help.

Updated figure.

L155: "3D printed plate" -> "3-dimensional (3D) printed plate"

Updated.

L158: Are these all the parameters?

Added frequency range to the list, this is technically a parameter that can be changed although since we've stated in the manuscript we are working with a certain bandwidth it was not included initially. Added for clarity.

L165: Same as the comment on line 135. Remove this link and place it in the data availability statement.

Removed.

L172: Something went wrong with the latex I imagine for the "χ_T". Double check the substricpt here.

Updated.

L174: Move the URL to the data availability statement here and elsewhere.

Updated.

Figure 2: This is probably just me misunderstanding something, but why is there a gap between the sensor and boat in panel D? Should something be connecting the two structures?

This was just an issue in image rendering – updated now.

Also, this figure caption is a tad confusing. I suggest describing each panel with its own sentence or two. For example, (A) is the ctd used …, etc. This makes it easier for people who are simply looking at the figure to obtain the necessary information quickly.

Updated.

Be consistent with your use of "3D" or "3-D"

Checked throughout.

L189: This sentence should be removed. There is no need to repeat what the section title already states.

Removed.

L191: I truly appreciate the authors' efforts to engage with an audience unfamiliar with underwater acoustics. As one of my main points in "Areas requiring clarification", please expand on more of the interpretation of the data here. There is hardly any discussion on how to interpret the figures provided, namely Figures 3 and 4.

See response to majoy comments avoid – we have expanded this section and the section on Figure 4 to help readers interpret echogram figures. Again, we thank the reviewer for pointing this out – very helpful comment from a non-acoustic audience member!

L211: "(<150-m)" Sometimes units are hyphenated and others are not. I would consider removing the hyphen here for consistency.

Updated all unit measurements throughout manuscript for consistency.

L221: In your list of objectives, the authors mention "geophysical parameter estimation" and now state "geophysical properties". What properties are these? Please explain here.

Updated this with some examples, also this section has been reworked so no appears in the introduction.

L223-227: "Broadband echosounder data collected in Hornsund fjord is here used to ..." is awkward phrasing. I'm not sure what describing the subsections here does either. Consider removing this for consolidation and summarizing it more in the introduction.

This suggestion has been implemented and moved to the introduction.

Figure 3: Similar to Figure 2, changing the caption to have dedicated sentences for each panel will make this caption much more readable. I am also confused about the units for panels A - D. Each panel should have it's own letter, too. "(SL" needs a closing parenthesis.

Updated with suggestions.

L362: "In high latitude glacial fjords this remote advantage" - > In high-latitude glacial fjords, this remote advantage. There is a missing hyphen and comma.

Updated, this section was moved up to the introduction.

L363: Since the authors are referring to general high-latitude glacial fjords and not a specific fjord, use an indefinite article "a" instead of "the" "such as the submerged ice-ocean interface of the glacial terminus" -> such as a submerged ice-ocean interface of a glacial terminus

Updated.

L364: "ice bergs" -> icebergs

Updated.

Figure 4: Maybe I am missing something, but what exactly is the dark blue region outlined by the dashed black (?) line? Please elaborate on this in the figure caption.

The figure caption noted the black dashed line highlights the terminus. We added arrows to the figure to point from the label in the figure "ice-ocean interface" to the scattering region.

L407: "strong backscattering return starting between 75 to 150 meters in range." The Y-axis only goes to a little past 120 m in range. Either correct this to 120 m or expand the Y-axis in Figure 4.

Updated.

L409: "peak intensity value of between -45 to -35 dB re one μPa." This unit is different than than those shown on Figure 4 (dB re 1/m) and Figure 3. Are the authors referring to panel D in Figure 5? Please clarify.

Updated the manuscript to remove these values.

L446-447: "With the proper deployment geometry, the broadband system  can provide direct observations of the subglacial discharge location; however, even" this sentence structure is repeated in this paragraph and used a lot in this manuscript. Consider rephrasing and varying this sentence structure for improved readability.

Updated.

L448-449: "backscattering signal from the plume higher in the water column can provide important observations related to plume dynamics." Where are the important observations the authors are referring to?

Updated.

L463 - 470: As stated in the major revisions, this paragraph is not necessary here. Consider either removing it or summarizing the main points and incorporating them into the discussion.

This has been updated.

L488: "tidal cycle (e.g.," The parentheses were never closed. Please add a closing parenthesis.

Updated.

L502: "(dark gray region, Fig. 5)" Add a panel to which part of Figure 5 you are referring to

Removed the reference to the "dark gray region"

Figure 5: Please increase the thickness of the profile lines in panels A and B. It is hard to distinguish colors with such thin lines.

Updated.

L543: "Taking the analysis of the acoustic measurements can be one step further". The wording of this sentence is off. Maybe "The analysis of the acoustic measurements can be expanded upon. For example…"

Great suggestion, implemented.

L564: "Challenges in applying acoustic inversion methods notwithstanding," This clause is awkward. Consider removing and starting the sentence with 'There is excellent…' The authors did a nice job of explaining the limitations of acoustic inversion above; no need to reiterate here.

Implemented

L577: Use future work instead of next step. This is more common and less colloquial.

Updated.

L585 - 600: Reintroducing broadband echosounders is very unnecessary this late in the manuscript. Again, from my major revision, this can either be removed or summarized and included in the introduction.

Removed this paragraph entirely.

L614 - 620: Same as above, this is not necessary here and better suited for the introduction.

We disagree, an in depth discussion of receiver sensitivity would slow the pace of the introduction and belongs in it's own section for those reader who are considering using these systems in the future. Outlining the importance of quality data collection requires it's own section.

However, we do understand this section, as originally written is overly long and we have tightened it up.

L640 - 670: I agree with Reviewer 1. I'm not exactly sure what this section adds. Maybe summarizing the main points of this section and adding it to 5.0 would suffice.

I see the main points as 1) accurate calibration and 2) acoustic absorption.

Tightened up this section.

L665 - 670: Why is this paragraph at the end of this section? Please consider moving this up in the section following the main points from above. This is the only instance where the authors explicitly mention post-processing. The order of this section in its current state is confusing. Explaining the important post-processing steps in Section 5 will succinctly summarize this section and reduce the paper length.

Moved this section up.

L675: "like those discussed in Sect. 5.2." Remove this and put (Section 5.2) and end the sentence.

Updated.

L707-710: The conclusions should be about the authors' work. It is unnecessary to discuss the history of the methods from other sources here.

We feel highlighting other potential analysis is important for a new audience, especially in the conclusion to reiterate that there are even more applications out there for these methods.

General comments about the conclusions: There were several references to future work in this manuscript. Placing that future work here is better for readers who skimmed the above sections. Either remove the future work references from the above sections and place them here, or briefly mention them again in this section.

Added.

---

## Author Comment (AC4)

High frequency broadband acoustic systems as a tool for high latitude glacial fjord research

Response to RC3

We have reviewed the suggestions made by RC3. We thank the reviewers for the detailed review of the paper and their suggestions, in particular the reviewer's concerns regarding deployment of broadband echosounders in proximity to actively calving glaciers. We appreciate the suggestion to make clear the safety consideration that must be made prior to collection in this hazardous region. Following their notes and suggestions, we have made changes in the manuscript using tracked changes and provided a clean version with all edits incorporated. We have responded to each comment/suggestion below (in red), noting where we incorporated the suggestions and noting the few instances where we disagreed with the suggestion and the reasoning behind our disagreement.

**Major Revision**

The claim that high-frequency broadband echosounders are a "low-cost, low-effort addition" to experimental field kits (line 19) significantly understates the logistical and safety challenges of operating near marine-termina ng glaciers. Actively calving glacier termini are extremely hazardous, and current safety guidelines typically recommend maintaining a minimum distance of at least 200–500 meters for crewed vessels depending on the glacier/location. The suggests on that this system could be routinely used in close proximity to the ice face without acknowledging these constraints is misleading and potentially dangerous. If the goal is to promote glacier-proximal observations, the authors should clearly state that such surveys must be conducted with uncrewed or remotely operated platforms to ensure safety. However, doing so would also require revising the argument about the system being low-cost and low-effort, since deploying autonomous vehicles in these environments is neither trivial nor inexpensive. This issue must be addressed directly to avoid mischaracterizing the feasibility of the method.

We appreciate the reviewer's perspective and suggestions on data collection in proximity to actively calving termini. Upon review of the manuscript, we can see why the reviewer is concerned we do not properly acknowledge safety considerations, particularly for very actively calving glaciers. We agree that safety should be the first consideration in planning field data collection.

We want to mention that for many glacial systems a minimum safety distance of 200 m is regularly used (as noted by the reviewer) and this does not preclude the use of ship-based deployment of these systems. A 200 kHz (center frequency) broadband scientific echosounder can operate up to 250 m range with reasonable signal to noise ratio and should the user be willing to drop the frequency range (for example to an ES120-7CD or ES70-7CD, both commercially available systems with the same price tag) the usable range increases to 450 and 750 m, respectively. While there are limitations that come with an increased range from the target, these system can collect the data described here without an uncrewed surface vehicle. We agree that remote surface vehicle deployment has the potential to improve data quality because of the possibility of closer deployment. But we believe that it is not necessary to state that one must use a remotely operated vehicle in all cases as it is very much dependent on the safety distance and the glacier in question.

However, we understand the concerns the reviewer has raised and have modified the manuscript to acknowledge the complexity (and safety concerns) related to working near glacial termini. To provide more context on the deployment potential of these systems we have added a section (5.4) on near-terminus deployments to our considerations section 5. Where we discuss the safety distance, range considerations, and potential for uncrewed

surface vehicle deployment. We reference this discussion in the introduction and section 4.2 – so that readers are aware of the need for safety considerations, the impact these considerations may have on data collection/processing/analysis and the limitation created by systems with very high ice cliffs and/or high calving rates. We do mention the possibility of using uncrewed surface vehicles and we acknowledge that incorporating a vehicle would bring up the cost of deploying these echosounders – should a user be starting their field kit from scratch.

**Minor Revisions**

The overall clarity and readability of the manuscript could be improved by more clearly distinguishing between background, methods, results, and interpretation. For instance, the sec on between lines 190-210, which provides helpful background on the echosounder system, might be better suited to a dedicated background or methods sec on rather than appearing in "Interpretation and analysis." As a reader, I would find it clearer if the results and interpretation were more distinctly separated, or if transitions between observation and analysis were made more explicit. Additionally, some content in lines 525–540 reads more like methodological detail and could be moved to the methods or an appendix. While these changes aren't strictly necessary, they would likely strengthen the manuscript's organization and make it easier to follow.

Based on feedback from all three reviewers we have significantly altered the structure of this paper – expanding the introduction, including the lines suggested here, to more broadly introduce echosounders as a tool for ocean science. We've also tightened up the analysis sections (section 4) and the discussion of deployment considerations (section 5). We believe these changes strengthen the manuscript's readability and thank the reviewer for their suggestions.

**Specific Line Edits**

Line 190–210: Move the echosounder background material into a dedicated methods or background section. It currently appears in "Interpretation and analysis," which is conceptually inconsistent.

As mentioned in above response, the manuscript has been significantly restructured, so this is now fixed.

Line 364: "ice bergs" → "icebergs"

Fixed.

Line 430: "Generally agreed…" lacks details → Were the extents the same size? Were they located in the same area? Quantifying this agreement would strengthen the claim (e.g., "Surface expression width was within X% of the width measured acoustically").

We have added to this sentence to clarify what basic agreements were observed – same position along the calving front and approximately the same surface expression (pool) extent as measured from time lapse time stamps and ship position compared against the acoustic observations.

Sec on 4.3: Consider moving this sec on before Sec on 4.2 to provide context on what geophysical parameters can be inferred from the acoustic signal before discussing plume/ice-face interpretations. I was curious as to what you think is scattering the signal within the plume a er reading section 4.3. Do you think it's sediment? Can you see that from the surface expression?

We feel that the discussion of broadband acoustic inversion is by far the most complex of all the analyses we discuss, which is why we left this section at the end. Since there is not a deep discussion of inversion efforts for the ice-face/plume data (and since we moved the introduction of inversion to the introduction of the paper), we feel the order of sections 4.1-4.3 should stay as is.

The mechanism responsible for the elevated scattering from the plume is still being investigated. This is a great question – we have not completed this analysis and so did not add in much beyond noting the possible scattering mechanisms. Given the scattering intensity and the frequency band in question, the scattering is likely not from suspended sediment, as the intensity is too high. There could be a component of suspended sediment adding to the overall scattering – we think it is likely that a combination of gas bubbles and intense mixing from the buoyant overturns are driving the scattering.

Line 489: Missing closing parenthesis after citation.

Fixed.

Line 525–540: This section mixes methodological description with results and interpretation. Consider moving parts of this to the methods sec on or appendix.

This section has been updated in the manuscript. The description and explanation of acoustic inversion (including previous examples) has been moved to the introduction of the paper. The analysis of the Hornsund fjord sill data remains in section 4.3, which does include method and results. This was purposeful, as this paper does not have a traditional methods section and separate results section. We have tightened section 4.3 significantly

and have moved some of the more technical acoustic equations and derivation to the appendix.

Line 565–574: Quantify agreement between model and observation ("within a factor of 2"?), and consider plotting predicted vs. modeled.

The agreement between the two measurements is stated on line 587-588 (updated manuscript). Figure 5 has been updated to provide both measurements in the time/space domain and the frequency domain.

**Figures**

Fig 3. Consider adding scale bars to panels showing KH instabilities

We have added scale bars to all four zoomed in panels (A-D).

Fig 4.

Can you clarify how SDP extent was determined? Was the outline based on a qualitative echogram interpretation, or was it mapped from surface ice mélange expression and transposed?

The plume was identified from the elevated scattering intensity; however, the identification of the plume on the image (outline) is not based on a quantitative metric such as an intensity threshold. Here, we are simply pointing out the ability of the system to observe the plume through elevated scattering – our current efforts (outside this manuscript) are focused on SGD plume analysis. That analysis will include a combination of the time lapse imagery and acoustic data. That analysis is beyond the scope of this paper. We noted in the figure caption the plume extent was manually mapped out based on elevated scattering intensity levels.

Is it possible that ambient plume signals appear in the echogram around ~250, ~525, or ~720 m along track? Can you comment on these signatures in the main text or caption?

It is possible – particularly at 350 m along track – the return between 50 and 70 m potentially contains individual bubbles. At 525 and 720 m along track the elevated scattering in proximity to the ice face is likely from sidelobe returns of an overhanging ice face – this was determined through split-aperture processes (not discussed in detail in this manuscript). These weakly scattering regions are noted in the main body of the manuscript in the paragraph that discusses the ice face return. We have updated the manuscript to note the positions of these returns to point readers towards this phenomenon and we've made note in the caption as well. The scattering around 350 m is more likely to be some

form of SGD – however, there was not a clear surface expression in this area and further analysis is needed before we could be certain to identify it – however, we did include a note in the caption as to the possibility that this is also discharge.

---

## Referee Report (RR1)

I feel that my major revision regarding safety considerations when collecting data near an actively calving marine-terminating glacier was appropriately addressed, and I appreciate that the authors have also improved the readability of the manuscript.

---

## Author Response (AR2)

Dear Dr Weidner and colleagues,

Thank you for your thorough revisions to the paper, and response to the reviewers. They find your changes acceptable and I am happy to move forward to publication. I still find the paper a little long and occasionally jumbled, so I have made the following suggestions to remove redundant sentences here and there. I suggest another thorough read to see if you can streamline any sections, but ultimately the scientific content is now understandable to interested readers.

L22: 'low effort' – I would caution that deploying anything in high latitudes can count as 'low effort', although you mean it relatively. Instead, could it be 'relatively straightforward', or 'simple to operate' or some other synonym?

L54: 'The aim of this manuscript is to outline the potential of broadband scientific echosounders to help answer many outstanding questions of these high-latitude coastal regions.' Could you be specific about the type of questions that can be answered? Obviously it is a broad, introductory statement, but I don't think this instrument can answer all questions about the high latitude coastal environments. Perhaps 'address questions about geophysical processes in glacier terminus environments' or something similar that narrows down the field.

Similarly, L56: add 'geophysical' or similar to 'improve *geophysical* observational capabilities in glaciated fjords'. Or combine these two sentences.

L65: 'remote nature of acoustic data collection' – is it truly remote? I think this sentence can go, as you say the same throughout the manuscript.

L84: move the comma to after 'system'

L231: 'the study site was' is redundant, begin this section with 'Hornsund fjord is...' and move the figure ref to the end of the sentence (after Svalbard).

L264: 'in Hornsund fjord' is redundant here. Combine the first two sentences of the para.

L365: 'during the Hornsund campaign' is redundant here. Remove.

L514: 'During the 2023 field campaign In Hansbukta' also redundant.

L546: I don't think these 'future efforts' fit well here. Suggest removing, or incorporating a brief sentence about them in the conclusions. There's a real jumble of results, interpretation and conjecture in this section, so maybe have a final read through and see if you can streamline.

L910: remove 'research efforts in Hornsund fjord discussed in this paper will be expanded upon in the future' – this is obvious, and the next two sentences are not really relevant to this paper. Instead, begin the para with L914 'Tracking the spatiotemporal variability of.....'

L920: 80k is not a relatively low start up cost in my budget! I would instead state the cost, without implied value judgement, and make this sentence concise: 'The start up cost is approximately 80k for transducer, receiver and license to operate, and many systems are portable with flexible deployment geometries'.

L928: remove 'all this points to' – sentence could simply read 'The application of broadband echo sounders in polar regions could thus enable an improved understanding of the complex geophysical dynamics of high latitude fjords'.

I look forward to the final version, and thanks for your submission to the Cryosphere and patience in obtaining reviewers with relevant expertise.

Liz

Editor, The Cryosphere.